# m⁶A epitranscriptomic modification regulates neural progenitor-to-glial cell transition in the retina

Yanling Xin, Qinghai He, Huilin Liang, Ke Zhang, Jingyi Guo, Qi Zhong, Dan Chen, Jinyan Li, Yizhi Liu, Shuyi Chen*

State Key Laboratory of Ophthalmology, Zhongshan Ophthalmic Center, Sun Yat-sen University, Guangdong Provincial Key Laboratory of Ophthalmology and Visual Science, Guangzhou, China

**Abstract** $N^6$-methyladenosine (m⁶A) is the most prevalent mRNA internal modification and has been shown to regulate the development, physiology, and pathology of various tissues. However, the functions of the m⁶A epitranscriptome in the visual system remain unclear. In this study, using a retina-specific conditional knockout mouse model, we show that retinas deficient in *Mettl3*, the core component of the m⁶A methyltransferase complex, exhibit structural and functional abnormalities beginning at the end of retinogenesis. Immunohistological and single-cell RNA sequencing (scRNA-seq) analyses of retinogenesis processes reveal that retinal progenitor cells (RPCs) and Müller glial cells are the two cell types primarily affected by *Mettl3* deficiency. Integrative analyses of scRNA-seq and MeRIP-seq data suggest that m⁶A fine-tunes the transcriptomic transition from RPCs to Müller cells by promoting the degradation of RPC transcripts, the disruption of which leads to abnormalities in late retinogenesis and likely compromises the glial functions of Müller cells. Overexpression of m⁶A-regulated RPC transcripts in late RPCs partially recapitulates the *Mettl3*-deficient retinal phenotype. Collectively, our study reveals an epitranscriptomic mechanism governing progenitor-to-glial cell transition during late retinogenesis, which is essential for the homeostasis of the mature retina. The mechanism revealed in this study might also apply to other nervous systems.

*For correspondence:
chenshy23@mail.sysu.edu.cn

Competing interest: The authors declare that no competing interests exist.

## Editor's evaluation

The authors' study investigated the role of m6A epitranscriptomic modification in the developing mouse retina. The study clearly demonstrated the defects of Mettl3CKO retina in mice, including cellular disorganization and abnormal physiological responses. Enriched scRNA-seq and MeRIP-seq data provide excellent resources to study the function of m6A modification in retinogenesis.

## Introduction

The retina is the neural sensory component of the visual system that is responsible for light perception and preliminary visual information processing. As a lateral derivative of the neural tube, the retina shares basic developmental, structural, and physiological principles with other neural tissues and has become an excellent model for studying neural biology. The retina is composed of six types of neurons, including rod and cone photoreceptors, bipolar cells, horizontal cells, amacrine cells, and retinal ganglion cells, that form complex neural circuits to detect and process visual signals (***Masland, 2001***). In addition to neurons, the retina contains one type of glial cell, Müller cells. As the sole glial cell inside the retina, Müller cells play pivotal roles in maintaining structural, physiological, and functional homeostasis in the neural retina (***Newman and Reichenbach, 1996***; ***Vecino et al., 2016***). During

**eLife digest** The retina is a layer in the eye that converts light into electrical signals, which allows us to see. It is a part of the central nervous system and is made of brain cells, such as neurons and supporting cells called glia. These supporting cells protect neurons, supply them with nutrients and maintain steady surrounding conditions. The retina shares many characteristics with other neural tissues, so it is useful for biologists to study these structures.

One way for cells to control the activity of genes is by chemically modifying messenger RNA molecules. These alterations can affect various aspects of mRNA and the proteins that are ultimately produced. The most common mRNA modification, referred to as m⁶A, plays a key role in the development and healthy performance of various tissues. However, it is unclear whether m⁶A is involved in how glial cells in the retina develop.

To address this question, Xin et al. studied the impact of blocking m⁶A in the retina of mice. These genetically modified mice displayed abnormalities as the retina developed. Analysis of the mRNA produced in single cells and the pattern of modifications revealed that m⁶A is involved in the development of glia. In particular, m⁶A helps to remove the mRNA associated with early-stage proto-glia, allowing the cells to mature and transition to their final form.

The finding by Xin et al. that the m⁶A RNA modification is an essential part of retina development could help to understand eye diseases. In addition, this discovery may apply to other brain regions, and, in time, such work could lead to new treatments for neurodegenerative diseases.

development, all retinal neurons and Müller cells are derived from multipotent retinal progenitor cells (RPCs), which go through a series of competent states to give rise to various types of retinal cells in a sequential while overlapping manner. For example, in mice, retinal ganglion cells, cone photoreceptors, and GABAergic amacrine cells are born early before birth, while glycinergic amacrine cells, rod photoreceptors, and bipolar cells are born late during the postnatal period before RPCs finally become Müller glial cells (*Agathocleous and Harris, 2009*; *Bassett and Wallace, 2012*; *Cepko, 2014*; *Heavner and Pevny, 2012*). Similar to glial cells in other neural tissues, Müller cells maintain significant levels of gene expression signatures of progenitors (*Blackshaw et al., 2004*; *Nelson et al., 2011*; *Roesch et al., 2008*), which has encouraged people to explore the in vivo reprogramming potentials of Müller cells for treating retinal degeneration disease purposes in recent years (*Hoang et al., 2020*; *Jorstad et al., 2017*; *Yao et al., 2018*; *Zhou et al., 2020*). However, to function as glial cells, Müller cells must develop and maintain their unique transcriptome distinct from that of RPCs (*Lin et al., 2019*; *Nelson et al., 2011*). How the transcriptome of Müller cells is established and maintained is currently not well understood.

RNA modifications constitute an important layer of post-transcriptional regulation that orchestrates the metabolism, location, and function of transcripts, which are collectively referred to as the epitranscriptome. Among the various modes of mRNA modification, *N*⁶-methyladenosine (m⁶A) is the most prevalent internal modification (*Meyer and Jaffrey, 2014*; *Meyer and Jaffrey, 2017*; *Roundtree et al., 2017*; *Shi et al., 2019*; *Zaccara et al., 2019*). m⁶A is cotranscriptionally installed on RNAs in the nucleus by a multisubunit methyltransferase 'writer' complex, in which METTL3 and METTL14 form the catalytic core; but only METTL3 has methyltransferase catalytic activity, while METTL14 functions as an allosteric activator (*Bokar et al., 1994*; *Bokar et al., 1997*; *Liu et al., 2014*; *Śledź and Jinek, 2016*; *Wang et al., 2016a*; *Wang et al., 2016b*). Two enzymes, FTO and ALKBH5, have been demonstrated to have m⁶A demethylase 'eraser' activity, emphasizing the reversible nature of the modification (*Jia et al., 2011*; *Zheng et al., 2013*). The effects of m⁶A on RNAs are mediated by various m⁶A readers, such as YTH family and HNRNP family proteins (*Lee et al., 2020*; *Zaccara et al., 2019*; *Zhao et al., 2017*). Gene knockdown and genetic knockout studies on m⁶A writers, erasers, and readers have shown that m⁶A plays important roles in regulating various aspects of organism development, physiology, and disease progression (*Frye and Blanco, 2016*; *Liu et al., 2019*; *Livneh et al., 2020*). However, the functions of the m⁶A epitranscriptome in the visual system remain unclear.

To test whether m⁶A plays roles during vertebrate retinal development, we used morpholinos to knock down the m⁶A-writer complex in zebrafish. Analysis of the retinal phenotype of these morphant zebrafish indicated that m⁶A promotes the timely differentiation and survival of RPCs in zebrafish

(*Huang et al., 2021*). In this study, using a mouse model of retina-specific conditional knockout of *Mettl3* and combining scRNA-seq, MeRIP-seq, and gene expression manipulation, we reveal that the m⁶A epitranscriptome plays critical roles during late retinogenesis by fine-tuning the transcriptomic transition from RPCs to Müller cells.

## Results
### *Mettl3* deficiency leads to structural and physiological abnormalities in the retina

To investigate the function of the m⁶A epitranscriptome in the retina, we conditionally knocked out *Mettl3*, the core component of the m⁶A-writer complex, in RPCs using *Six3-Cre* (*Furuta et al., 2000*) and *Mettl3$^{floxed}$* (*Lin et al., 2017*) mice. METTL3 is abundantly and ubiquitously expressed in the retina from early embryonic age to adult stage (*Figure 1—figure supplement 1A–D*). Immunofluorescence staining (IF) demonstrated that *Mettl3* was efficiently knocked out in the central retinas of *Six3-Cre$^{+}$*; *Mettl3$^{floxed/floxed}$* mice (hereafter referred to as *Mettl3*-CKO) (*Figure 1A and A'*), and MeRIP-qPCR examination of several candidate genes showed that m⁶A levels were efficiently downregulated in *Mettl3*-CKO retinas (*Figure 1—figure supplement 1E*). Even though *Mettl3* was efficiently deleted from RPCs, the retinas of *Mettl3*-CKO mice at p0 appeared grossly normal, exhibiting an already separated retinal ganglion cell layer (RGL) and a still-developing retinoblast layer (RBL), comparable to littermate control retinas (*Figure 1B and B'*). However, at p14, the time point when retinogenesis should have completed and mature retinal organization should have been established, histological examination showed that the *Mettl3*-CKO retinas exhibited severe structural disorganization (*Figure 1C and C'*). At this age, the control retinas were well laminated into three cellular layers and two synaptic layers (*Figure 1C*). In the mutant retinas, although all the cellular and synaptic layers were distinguishable, the outer nuclear layer (ONL) presented many rosettes, and some inner nuclear layer (INL) cells were drawn into the ONL (*Figure 1C'*, the asterisks indicate the rosette structures). We then examined the visual function of the retina using multifocal electroretinogram (mfERG). The results showed that the visual responses from most regions of *Mettl3*-CKO retinas exhibited abnormal wave forms and much reduced P1 peak amplitudes compared with control retinas (*Figure 1D*), demonstrating disrupted visual function of the mutant retinas. To examine whether retinal neurons and glial cells were generated in structurally distorted *Mettl3*-CKO retinas, we performed IF analyses of characteristic markers of various types of retinal cells on p15 retinal sections. The IF results showed that all types of retinal cells, including RECOVERIN$^{+}$ photoreceptors, VSX2$^{+}$ bipolar cells, SOX9$^{+}$ Müller glia, CALBINDIN$^{+}$ horizontal cells, HPC-1$^{+}$ amacrine cells, and BRN3A$^{+}$ retinal ganglion cells, were generated in *Mettl3*-CKO retinas, although some cells in the INL, especially bipolar cells, Müller glia, and horizontal cells, were displaced into the ONL (*Figure 1—figure supplement 2*).

In the retina, Müller cells monitor the retina by extending elaborate cellular processes to interact with every retinal neuron and form an outer limiting membrane (OLM) and an inner limiting membrane (ILM) to ensheath the entire retina (*Wang et al., 2017*). Mechanical and genetic ablation studies have demonstrated essential roles of Müller cells in maintaining the structural and functional homeostasis of the retina (*Byrne et al., 2013*; *MacDonald et al., 2015*; *Shen et al., 2012*). To examine how Müller cells were affected by *Mettl3* knockout, we placed *Mettl3*-CKO mice on the *Rlbp1-GFP* mouse background, whose Müller glia express GFP (*Vázquez-Chona et al., 2009*). GFP IF illustrated severely distorted morphology of Müller glia in *Mettl3*-CKO retinas: somata of many mutant Müller glia translocated from their normal position in the middle of the INL to move upward and mingle with photoreceptors in the ONL (*Figure 1E'*, the asterisks indicate the soma of Müller glia). Moreover, the apical edges of the mutant Müller glia failed to reach the apical edge of the retina, where they normally organize into the OLM; instead, they formed a discontinuous OLM that was separated into each rosette structure (*Figure 1E'*, the white arrows indicate the broken OLM). We then further examined the OLM by performing N-cadherin IF, which showed that the OLM in *Mettl3*-CKO retinas became discontinuous and exhibited many breaks, through which many photoreceptors fled into the subretinal space (*Figure 1F'*, the white arrows indicate breaks in the OLM). By examining the retinas of pups at different postnatal ages, we found that occasional breaks along the OLM started to appear as early as p6 (*Figure 1G'*, the white arrowheads indicate breaks in the OLM), coinciding with the time when morphological maturation of Müller glia starts (*Wang et al., 2017*).

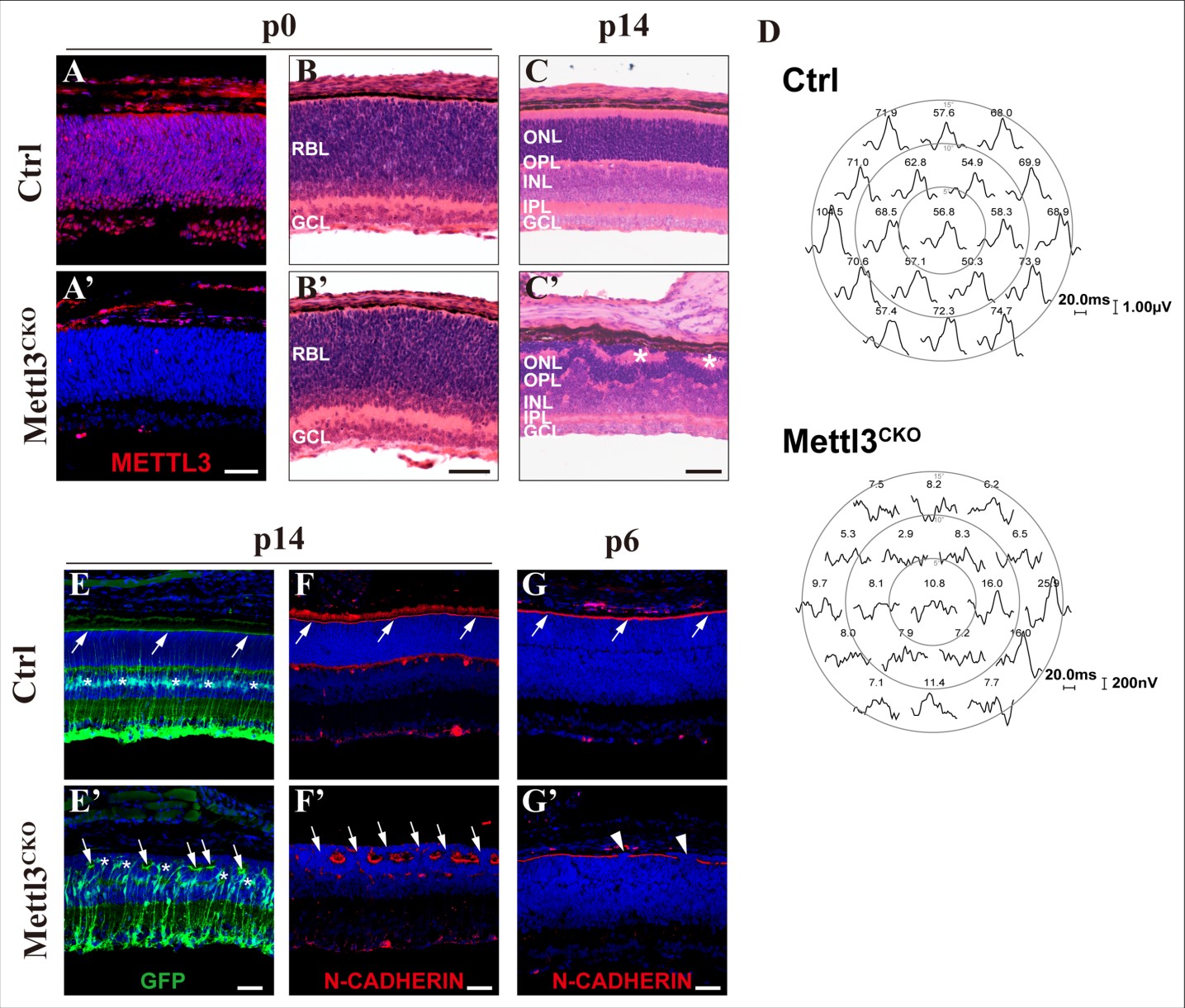

**Figure 1.** *Mettl3* deficiency leads to structural and physiological abnormalities in the retina. (**A, A'**) Confocal images of METTL3 expression in p0 control (Ctrl) and *Mettl3*-CKO retinas showing that *Mettl3* was efficiently deleted in the mutant retinas. (**B, B'**) H&E staining of p0 control and *Mettl3*-CKO retinas showing that the mutant retinas had a grossly normal histological structure. (**C, C'**) H&E staining of p14 control and *Mettl3*-CKO retinas showing that the mutant retinal structure was severely disrupted. * indicates rosette structures. (**D**) Multifocal electroretinogram (ERG) response recordings in each hexagonal region of the retinas stimulated with 19 hexagonal light signals showing that the visual function of *Mettl3*-CKO retinas was severely disrupted. The numbers on the top of each waveform plot indicate the averaged ERG response in the respective hexagonal area in nV/deg$^2$. (**E, E'**) Confocal images of p14 retinas labeled with GFP to highlight Müller cells (mice on the *Rlbp1-GFP* background). The white arrows indicate the outer limiting membrane (OLM). * in (**E'**) indicates mislocalized somata of Müller cells. (**F, F'**) Confocal images of p14 retinas stained for N-CADHERIN to highlight the OLM. The white arrows in (**F**) indicate the OLM, and those in (**F'**) indicate breaks between the OLM. (**G, G'**) Confocal images of p6 retinas stained for N-CADHERIN showing that breaks in the OLM (white arrowheads) in *Mettl3*-CKO retinas started to appear at p6. RBL: retinoblast layer; GCL: ganglion cell layer; ONL: outer nuclear layer; OPL: outer plexiform layer; INL: inner nuclear layer; IPL: inner plexiform layer. Scale bars are 50 μm.

The online version of this article includes the following figure supplement(s) for figure 1:

**Figure supplement 1.** The expression of *Mettl3* in the mouse retina.

**Figure supplement 2.** All types of retinal cells are generated in *Mettl3*-CKO retinas.

Taken together, these results demonstrate that *Mettl3* deficiency leads to structural and functional abnormalities in the mature retina and indicate that dysfunction of Müller glial cells might be the cause of these abnormalities.

### *Mettl3* deficiency distorts late-stage retinogenesis

We next performed BrdU labeling assays to examine the proliferative activity of RPCs. At E15.5, there were similar numbers of BrdU⁺ cells in *Mettl3*-CKO retinas and control retinas (**Figure 2—figure**

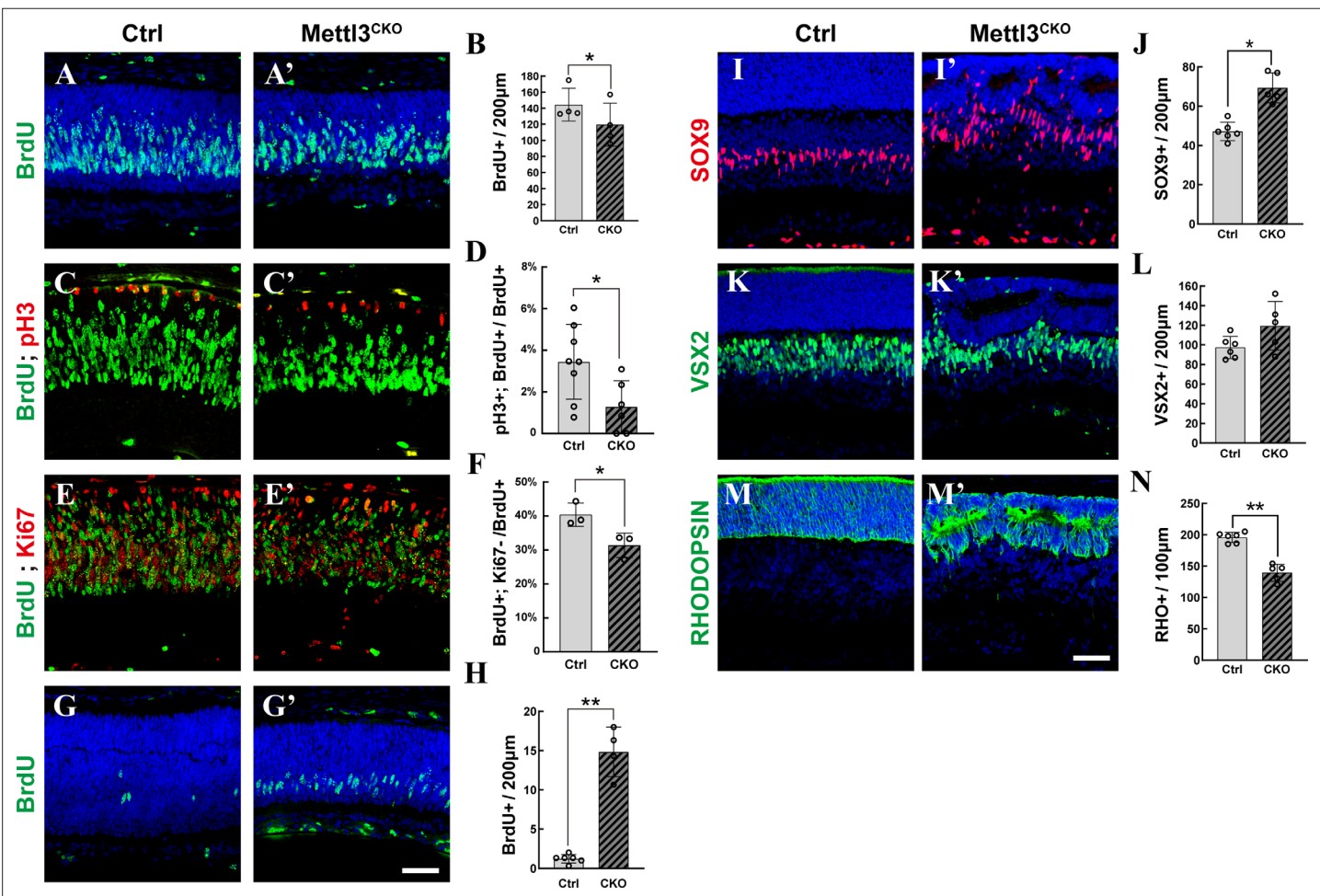

**Figure 2.** *Mettl3* deficiency distorts late-stage retinogenesis. (**A, A'**) Confocal images of p1 retinas stained for BrdU. The pups were injected with BrdU, and their retinas were collected 2 hr later. (**B**) Quantification of the BrdU⁺ cells in (**A**) and (**A'**). (**C, C'**) Confocal images of p1 retinas stained for BrdU and pH3. The pups were injected with BrdU, and their retinas were collected 6 hr later. (**D**) Quantification of the BrdU⁺ and pH3⁺ cells in (**C**) and (**C'**). (**E, E'**) Confocal images of p3 retinas stained for BrdU and KI67. The pups were injected with BrdU at p1, and their retinas were collected 48 hr later. (**F**) Quantification of the BrdU⁺ and Ki67⁻ cells in (**E**) and (**E'**). (**G, G'**) Confocal images of p5 retinas stained for BrdU. The pups were injected with BrdU, and their retinas were collected 2 hr later. (**H**) Quantification of the BrdU⁺ cells in (**G**) and (**G'**). (**I, I'**) Confocal images of p7 retinas stained for SOX9 to illustrate Müller cells. (**J**) Quantification of the SOX9⁺ cells in (**I**) and (**I'**). SOX9⁺ cells in the retinal ganglion cell layer and inner plexiform layer were excluded to avoid miscounting of SOX9⁺ astrocytes. (**K, K'**) Confocal images of p7 retinas stained for VSX2 to illustrate bipolar cells. (**L**) Quantification of the VSX2⁺ cells in (**K**) and (**K'**). (**M, M'**) Confocal images of p7 retinas stained for RHODOPSIN to illustrate rods. (**N**) Quantification of the RHODOPSIN⁺ cells in (**M**) and (**M'**). The data in (**B, D, F, H, J, L, N**) are presented as the means ± standard deviations, corresponding to at least three independent biological replicates. * p<0.05, **p<0.01. The scale bars in (**G'**) and (**M'**) are 50 µm and apply to (**A–G'**) and (**I–M'**), respectively.

The online version of this article includes the following figure supplement(s) for figure 2:

**Figure supplement 1.** *Mettl3* deficiency does not affect the proliferation of early retinal progenitor cells (RPCs).

**Figure supplement 2.** *Mettl3* is required in late retinal progenitor cells (RPCs) for the proper cell cycle progression.

**Figure supplement 3.** Different types of retinal cells at p7.

**Figure supplement 4.** Retinas deficient for *Mettl3* gradually degenerate due to apoptosis.

*supplement 1*), suggesting that *Mettl3* deficiency does not affect the proliferation of early RPCs. However, at p1, fewer BrdU⁺ cells were observed in *Mettl3*-CKO retinas than in control retinas (***Figure 2A, A' , and B***), indicating that there were fewer proliferating RPCs in *Mettl3*-CKO retinas. To examine whether the cell cycle progression of late RPCs was affected by *Mettl3* deletion, we harvested p1 retinas at 6 hr after BrdU injection and subjected them to costaining with BrdU and pH3. We calculated the percentage of BrdU⁺; pH3⁺ RPCs vs. BrdU⁺ RPCs, which showed that the percentage was significantly lower in *Mettl3*-CKO retinas than in control retinas (***Figure 2C, C' , and D***), suggesting that S-to-M phase progression was prolonged in *Mettl3*-CKO late RPCs. The postnatal period occurs at the end of retinogenesis, when an increasing number of RPCs withdraw from the cell cycle and undergo terminal differentiation. We examined the rate of cell cycle withdrawal of RPCs by injecting BrdU at p1 and monitored their KI67 expression 48 hr later at p3. The results showed that there were significantly fewer BrdU⁺; KI67⁻ RPCs in *Mettl3*-CKO retinas at p3 (***Figure 2E, E' , and F***), meaning that *Mettl3*-CKO late RPCs withdrew from the cell cycle slower than control cells. Consistent with this slower withdrawal rate, at p5, when proliferating retinogenesis events were mostly terminated in control central retinas (***Figure 2G and H***), a significant number of proliferating cells were still observable in *Mettl3*-CKO central retinas (***Figure 2G' and H***). To test the role of *Mettl3* in late RPCs directly, we performed in vivo electroporation on retinas of p1 pups to specifically introduce shRNAs targeting *Mettl3* into late RPCs. We harvested the retinas at p5 after 2 hr of BrdU labeling. Because our in vivo electroporation procedure often targets peripheral region of the retina where retinogenesis continues at this time point, we still observed many BrdU⁺ cells in the region, but most of these BrdU⁺ cells were GFP⁻ in the retinas electroporated with control plasmids (***Figure 2—figure supplement 2A and B***). However, a significantly higher percentage of GFP⁺ cells were co-labeled with BrdU in the retinas electroporated with plasmids expressing shRNAs targeting *Mettl3* (***Figure 2—figure supplement 2A' and B***), suggesting that cell cycle withdrawal was delayed in the experimental group. Taken together, these experiments demonstrated that *Mettl3* is required in late RPCs for their proper progression through the cell cycle and timely withdrawal from retinogenesis. This end stage of retinogenesis is the peak time point at which rods, bipolar cells, and Müller glia are generated. To examine whether *Mettl3* deficiency affects retinal differentiation, we counted the numbers of rods, Müller glia, and bipolar cells in the retina at p7, when the major retinogenesis phase has passed in the central retinas, even in *Mettl3*-CKO mice. The results showed that the number of SOX9⁺ Müller glia was increased in *Mettl3*-CKO retinas (***Figure 2I, I' , and J***, SOX9⁺ cells in the ganglion cell layer and inner plexiform layer were excluded to avoid miscounting astrocytes). On the other hand, the number of VSX2⁺ bipolar cells remained relatively unchanged (***Figure 2K, K' , and L***), while the number of RHODOPSIN⁺ photoreceptors was reduced (***Figure 2M, M' , and N***). We also determined the cell densities of other retinal neurons. The number of BRN3A⁺ RGCs, CALBINDIN⁺ amacrine cells, and CAR⁺ cones was comparable between mutant and control retinas (***Figure 2—figure supplement 3A–F***), while the number of CALBINDIN⁺ horizontal cells was significantly reduced in the mutant retinas (***Figure 2—figure supplement 3E, E', and G***).

To examine whether *Mettl3* deficiency affects the survival of retinal cells, we performed TUNEL assay, which showed that apoptosis was significantly elevated in *Mettl3*-CKO retinas (***Figure 2—figure supplement 4A–D***). Costaining with cell-type markers showed that both RPCs and their differentiated progenies underwent apoptosis (***Figure 2—figure supplement 4E–G***). The elevation in cell death events resulted in severe degeneration of all cellular layers in *Mettl3*-CKO retinas at older ages (***Figure 2—figure supplement 4H and H'***) and might have contributed to the reduced population of photoreceptors at p7 and the reduced population of BrdU⁺ cells at p0.

## Single-cell transcriptome analyses demonstrate distorted and extended late-stage retinogenesis in *Mettl3*-CKO retinas

To examine how *Mettl3* deficiency affects gene expression in the retina, we performed RNA-seq analyses of control and *Mettl3*-CKO retinas at p7, the time point at which the structural defects became obvious. In principal component analysis (PCA), mutant retinas were clearly separated from control retinas (***Figure 3—figure supplement 1A***). Differentially expressed gene (DEG) analysis showed that 580 genes were upregulated and 532 genes were downregulated in *Mettl3*-CKO retinas (***Figure 3—figure supplement 1B***). Pearson analysis showed only subtle differences between control and mutant

retinal transcriptomes (*Figure 3—figure supplement 1C*), indicating that *Mettl3* delicately regulates retinal gene expression.

To examine the transcriptomic and cell status changes in *Mettl3*-CKO retinal cells in a more comprehensive way, we performed single-cell RNA sequencing (scRNA-seq) on control and *Mettl3*-CKO retinal cells at p7. A total of 5995 control and 6992 *Mettl3*-CKO retinal cells passed quality control and were used for subsequent bioinformatic analysis. By applying unsupervised clustering and dimensional reduction and based on the expression of key marker genes, sequenced cells were clustered into nine clusters, representing the typical cellular composition of the retina at p7 (*Figure 3A and B*, *Figure 3—figure supplement 2A*). Rods, the largest cell population in the retina, occupied the largest cluster, and other retinal neurons, including cones, amacrine cells, bipolar cells, and Müller glial cells, each formed distinct clusters. Retinal ganglion cells and horizontal cells were not represented, probably due to their small population size. A group of cells along the trajectory toward bipolar cells and rods were judged as precursors for the two cell types based on their high expression of *Otx2* and *Neurod4*. A group of cells at one edge extending from Müller glia highly expressed cell cycle drivers, such as *Mki67*, *Pcna,* and *Ccnb1*, suggesting that they were RPCs and were likely RPCs in the final phase of differentiation that reside on the retinal periphery at this time point; these cells were more abundantly observed in *Mettl3*-CKO retinas whose retinogenesis period was extended (*Figure 3— figure supplement 3*).

Cell number counting showed that the proportion of rods was slightly reduced in *Mettl3*-CKO retinas; the proportion of RPCs in *Mettl3*-CKO retinas was nearly twofold higher than that in control retinas (*Figure 3D*, *Figure 3—figure supplement 2B*), consistent with the extended retinogenesis observed for *Mettl3*-CKO RPCs. Our manual counting showed that more Müller glial cells were generated in *Mettl3*-CKO retinas; however, the scRNA-seq results showed that the proportions of Müller cells in control and *Mettl3*-CKO retinas were comparable (*Figure 3D*, *Figure 3—figure supplement 2B*). These seemingly conflicting results were due to the different sampling strategies adopted in the two experiments. For manual counting, we selected the central region of the retina, while for scRNA-seq, we collected the whole retina. At p7, the remaining KI67$^+$ retinogenesis region was larger in *Mettl3*-CKO retinas than in control retinas, thus, the region of the retina that had finished Müller gliogenesis was smaller in *Mettl3*-CKO retinas than in control retinas (*Figure 3—figure supplement 3*). As the consequence, the elevated Müller gliogenesis phenotype was masked by the smaller region that had finished the process in the scRNA-seq data analysis. To observe the progression of this end-stage retinogenesis, we selected RPCs and Müller glia for pseudotime analysis. The results demonstrated a continuous cell status transition from RPCs to Müller glia (*Figure 3E*), consistent with the gliogenic status of RPCs at this time point. Analysis of the pseudotime distributions of control and *Mettl3*-CKO cells revealed that the gliogenesis progression of *Mettl3*-CKO RPCs lagged behind that of control RPCs (*Figure 3F*), suggesting delayed Müller gliogenesis in *Mettl3*-CKO retinas, which is consistent with the extended retinogenesis in *Mettl3*-CKO retinas revealed by BrdU labeling (*Figure 2*). Thus, the scRNA-seq examination of p7 control and *Mettl3*-CKO retinal cells reveals distorted and extended late-stage retinogenesis in the mutant retina at the transcriptomic level.

## *Mettl3* deficiency distorts the transcriptomes of late RPCs and Müller glial cells

Clustering analysis of all the cells sequenced failed to reveal distinct cell clusters for the cells derived from the mutant retinas (*Figure 3C*). We suspected that subtle transcriptomic changes in some types of retinal cells were masked in the whole dataset. We thus performed reclustering of individual clusters of cells. The reclustering revealed distinct mutant clusters in the Müller, RPC, bipolar, and rod populations (*Figure 3G and K*, *Figure 3—figure supplement 4A*), especially for the Müller glia population, in which the mutant cell cluster was distantly separated from the control cell cluster (*Figure 3K*). Cell cycle phase analysis of RPCs showed that G2/M and S phase cells were distinctively distributed along two sides of UMAP, but the cell cycle phases were not responsible for the formation of the mutant cell cluster (*Figure 3—figure supplement 4B*). Since phenotypic analyses indicated that RPCs and Müller glia were the two cell types most significantly affected by *Mettl3* deletion and that the mutant cell clusters were more pronounced in the RPC and Müller populations, we next focused on RPCs and Müller glia. DEG analysis of the mutant RPC cell cluster and the remaining RPC cells showed that 286 genes were upregulated and 212 genes were downregulated in mutant RPCs (*Figure 3H* and *Source*

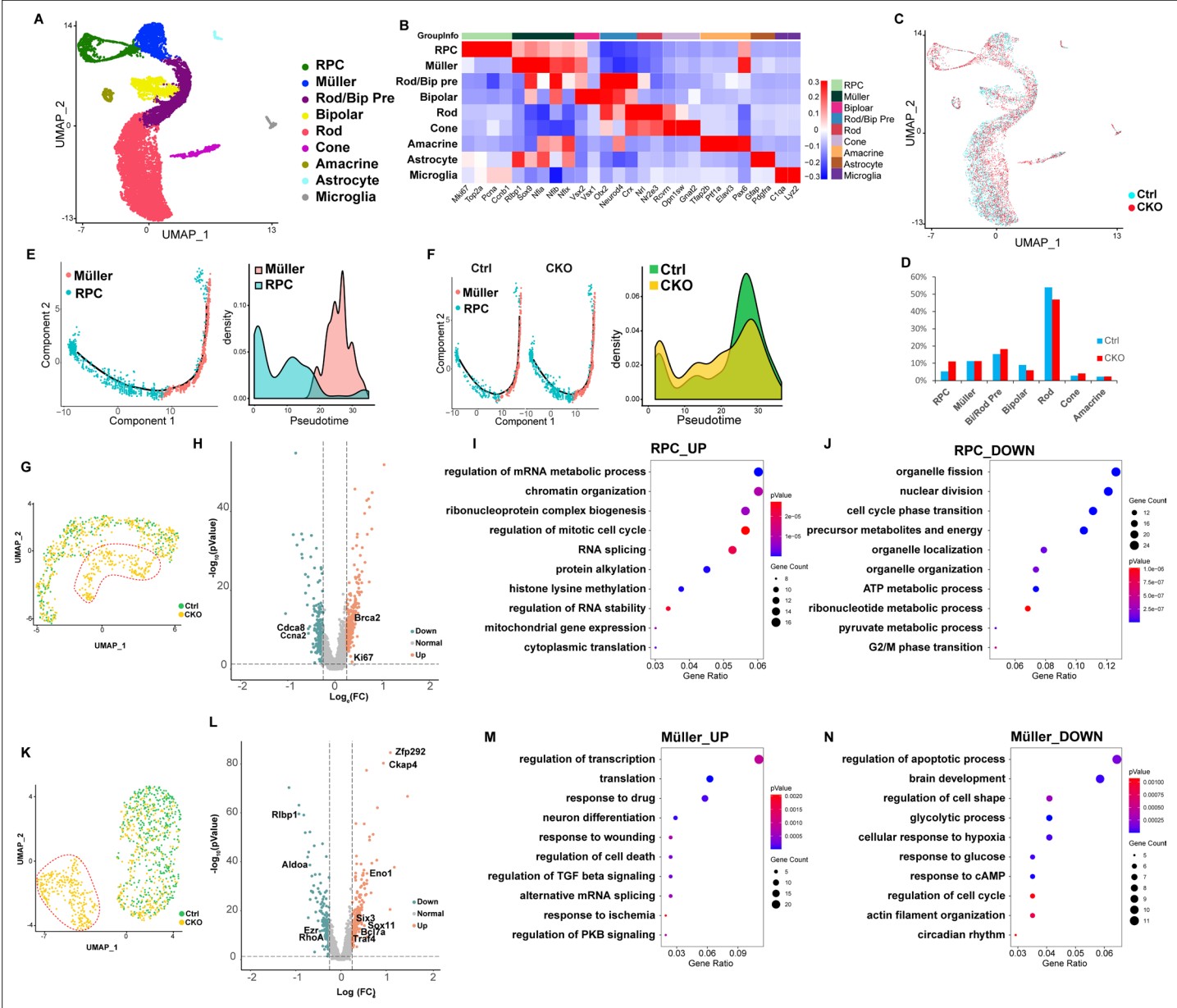

**Figure 3.** Single-cell RNA sequencing (scRNA-seq) analyses reveal distorted transcriptomes of late retinal progenitor cells (RPCs) and Müller glial cells in *Mettl3*-CKO retinas. (**A**) UMAP presenting the cell clusters of all sequenced cells. (**B**) Heatmap illustrating the expression patterns of key markers of each retinal cell type. (**C**) UMAP presenting the distribution of control and *Mettl3*-CKO retinal cells. (**D**) Bar column presenting the cell compositions in control and *Mettl3*-CKO retinas. (**E**) Left: pseudotime presentation illustrating the developmental progression from RPCs to Müller glia; right: cell density distributions of RPCs and Müller glia over the pseudotime period. (**F**) Left: separate pseudotime presentations of control and *Mettl3*-CKO RPCs and Müller glia; right: cell density distributions of control and *Mettl3*-CKO cells over the pseudotime period. (**G**) UMAP of reclustered RPCs. The red dotted line encircles a distinct cell cluster composed solely of *Mettl3*-CKO RPCs. (**H**) Volcano plot showing gene expression differences between mutant RPCs (encircled by the red dotted line in **A**) and the remaining RPCs. (**I**) Bubble plot showing the biological processes enriched in genes that were upregulated in the *Mettl3*-mutant RPC cluster. (**J**) Bubble plot showing the biological processes enriched in genes that were downregulated in the *Mettl3*-mutant RPC cluster. (**K**) UMAP of reclustered Müller glia. The red dotted line encircles a distinct cell cluster composed solely of *Mettl3*-CKO Müller glia. (**L**) Volcano plot showing gene expression differences between mutant Müller glia (encircled by the red dotted line in **E**) and the remaining Müller glia. (**M**) Bubble plot showing the biological processes enriched in genes that were upregulated in the *Mettl3*-mutant Müller glial cluster. (**N**) Bubble plot showing the biological processes enriched in genes that were downregulated in the *Mettl3*-mutant Müller glial cluster.

The online version of this article includes the following figure supplement(s) for figure 3:

**Figure supplement 1.** RNA-seq analyses of the transcriptomes of control and *Mettl3*-CKO retinas.

**Figure supplement 2.** Single-cell RNA sequencing (scRNA-seq) analysis of *Mettl3*-CKO retinal cells.

*Figure 3 continued on next page*

**Figure supplement 3.** Proliferating retinal progenitor cells (RPCs) in the retinas at p7.

**Figure supplement 4.** Reclustering of different types of retinal cells.

*data 1*). The upregulated genes were enriched for biological processes related to gene expression regulation, while downregulated genes were enriched for biological processes related to organelle, metabolism, and cell cycle regulation (*Figure 3I and J*). Importantly, key cell cycle machinery components, such as *Ki67* and *Brca2*, were upregulated, but many cell cycle regulators, such as *Ccna2* and *Cdca8*, were downregulated (*Figure 3H–J* and *Source data 1*), which might be a part of the molecular mechanism underlying the extended cell cycle progression observed in *Mettl3*-CKO RPCs (*Figure 2*). DEG analysis of the mutant Müller cell cluster and the remaining Müller cells showed that 239 genes were upregulated and 181 genes were downregulated in mutant Müller cells (*Figure 3L* and *Source data 2*). The upregulated genes in the mutant Müller cell cluster were enriched for genes regulating transcription that have been linked with RPC development, such as *Six3* and *Sox11* (*Figure 3L and M* and *Source data 2*). The downregulated genes in the mutant Müller cell cluster were enriched for regulators participating in 'cell shape' and 'actin filament organization,' including important actin cytoskeleton regulators, such as *RhoA* and *Ezr* (*Figure 3L and N* and *Source data 2*), indicating compromised physical properties of mutant Müller cells, which explained the distorted morphology of Müller cells and severely disorganized tissue structure in *Mettl3*-CKO retinas (*Figure 1*). In addition, many genes that are essential for the supporting functions of Müller cells, such as those involved in the 'glycolytic process,' including *Aldoa*, *Eno1*, and *Rlbp1*, were significantly downregulated in the mutant Müller cell cluster (*Figure 3L and N* and *Source data 2*), indicating that the physiological functions of Müller glia in *Mettl3*-CKO retinas were compromised, which likely contributes to the functional abnormality of *Mettl3*-CKO retinas.

Taken together, reclustering and DEG analyses of scRNA-seq data reveal that *Mettl3* deficiency distorts the transcriptomes of late RPCs and Müller cells, which is likely the molecular basis for the distorted cellular behaviors of RPCs and Müller glia in *Mettl3*-CKO retinas.

## The m⁶A epitranscriptome landscape of the mouse retina

To explore how $m^6A$ modification might be involved in retinal development and physiology, we next characterized the mouse retinal $m^6A$ epitranscriptome using MeRIP-seq. With a fold enrichment cutoff of 3, we detected 15,471 peaks on 6735 genes, 11,601 peaks on 6356 genes, and 14,041 peaks on 5940 genes in p0, p7, and adult retinas, respectively (*Figure 4A* and *Source data 3*). The retinal $m^6A$ epitranscriptomes at different ages were similar to each other that of the 8772 $m^6A$-modified genes detected in the retina, 46% were detected in all ages, and 71% were detected in at least two ages (*Figure 4A*). The retinal $m^6A$ peaks were largely localized in the CDS region, with pronounced enrichment around the stop codon (*Figure 4B*, *Figure 4—figure supplement 1A*), and enriched for the 'GGACU' sequence (*Figure 4C*), consistent with the topological features observed in the $m^6A$ epitranscriptomes of other tissues (*Dominissini et al., 2012*; *Meyer et al., 2012*; *Zaccara et al., 2019*; *Zhao et al., 2017*). The mouse retinal $m^6A$ epitranscriptome was enriched for genes participating in biological processes related to retinal development, structural organization, and visual function, such as 'synapse organization,' 'axonogenesis,' 'dendrite development,' 'cell junction assembly,' and 'retina development,' including many genes that have been demonstrated to play essential roles in retinal development, physiology, or pathology (*Figure 4D and E*, *Figure 4—figure supplement 1B*). The retinal $m^6A$ epitranscriptome at p0 was further enriched for genes involved in cell cycle progression, consistent with the proliferative status of the tissue at the time (*Figure 4—figure supplement 1B* and *Source data 4*). Thus, the MeRIP-seq analyses revealed that the retinal $m^6A$ epitranscriptome is enriched for genes that play key roles in many essential activities of the retina and might participate in regulating these biological processes.

## m⁶A modification promotes the degradation of RPC transcripts during RPC-to-Müller transition

To investigate how $m^6A$ modification regulates the retinal transcriptome, we performed integrative analyses of the scRNA-seq data and the MeRIP-seq data. First, we analyzed the expression levels

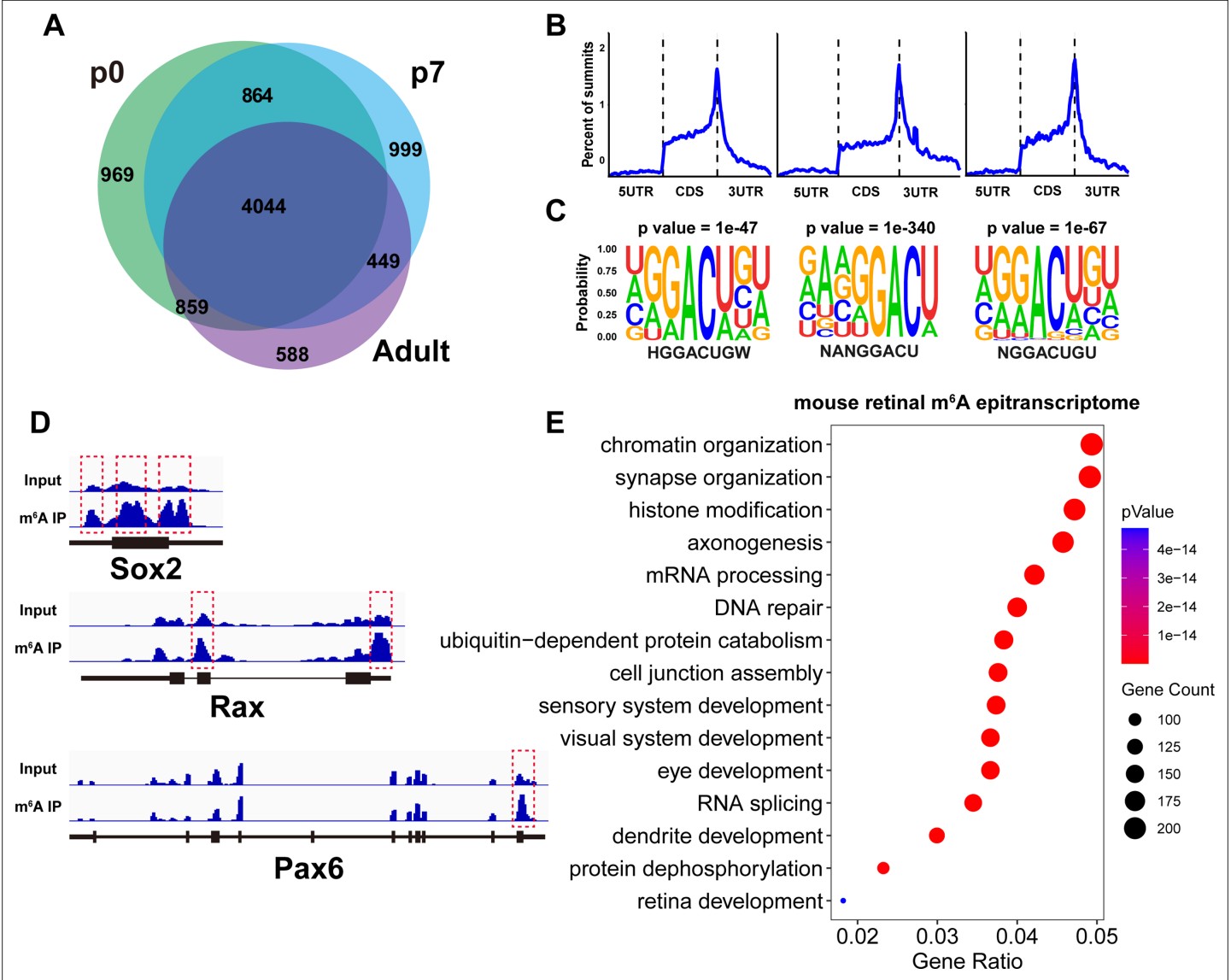

**Figure 4.** Mouse retinal m⁶A epitranscriptome. (**A**) Pie graph illustrating the number of genes carrying m⁶A modification detected by MeRIP-seq in the mouse retina at different ages. The numbers represent the number of genes in the respective pie region. (**B**) Plots illustrating the distribution of m⁶A peaks along mouse retinal transcripts. (**C**) The most enriched motif among m⁶A peak sequences in the mouse retina transcriptome. (**D**) IGV views illustrating the m⁶A peak distribution along transcripts of important retinal regulatory genes revealed by MeRIP-seq. (**E**) Bubble plot showing the biological processes enriched in genes carrying m⁶A modification detected in the retinas of mice at all ages.

The online version of this article includes the following figure supplement(s) for figure 4:

**Figure supplement 1.** m⁶A epitranscriptomes of the mouse retina at different ages.

of m⁶A-tagged retinal transcripts in control and *Mettl3*-mutant cell clusters. Single-sample gene set enrichment analysis (ssGSEA) showed that m⁶A-tagged transcripts tended to be upregulated in *Mettl3*-mutant RPC and Müller glia clusters, and in *Mettl3*-mutant rod and bipolar cell clusters, but to a lesser extent (*Figure 5A*). Grouping genes based on the number of m⁶A peaks showed that as the number of m⁶A sites increased, the genes were upregulated to greater levels in *Mettl3*-mutant RPC and Müller glia clusters (*Figure 5B*). These results suggest that m⁶A modification promotes the degradation of modified transcripts in RPCs and Müller cells. Examining the m⁶A modification status of the DEGs in *Mettl3*-mutant RPC and Müller glia clusters showed that most of the upregulated genes in *Mettl3*-mutant clusters were subject to m⁶A modification, while, in contrast, a much smaller portion of the downregulated genes did (*Figure 5C*), suggesting that one major direct cause for

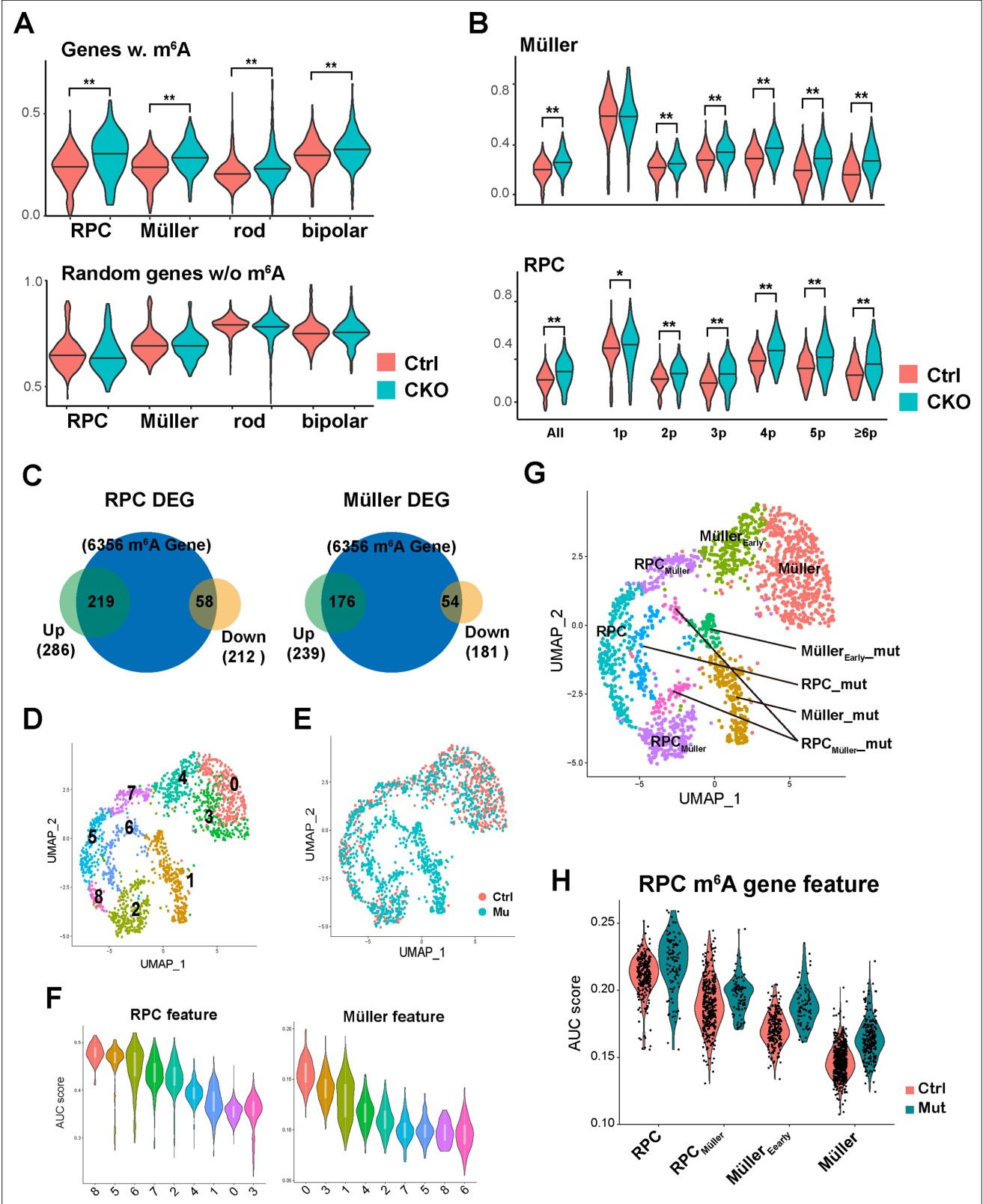

**Figure 5.** m⁶A modification promotes the degradation of retinal progenitor cell (RPC) transcripts. (**A**) Top: violin plot of single-sample gene set enrichment analysis (ssGSEA) scores illustrating the expression levels of m⁶A-modified genes in control and *Mettl3*-CKO retinal cells. Bottom: violin graph of ssGSEA scores of randomly selected genes that do not carry m⁶A to serve as controls. (**B**) Violin plots of ssGSEA scores illustrating the expression levels of different groups of m⁶A-modified genes in control and *Mettl3*-CKO Müller glia (top) and RPCs (bottom). The genes are grouped

*Figure 5 continued*

based on the number of m⁶A peaks along their transcript. (**C**) Pie graphs showing the number of m⁶A-modified genes that were upregulated (green pies) or downregulated (yellow pie) in *Mettl3*-mutant RPC or Müller glia clusters. The numbers inside the parentheses indicate the number of genes, and the numbers inside the figure indicate the number of overlapping genes between the two pies. (**D**) UMAP of reclustered RPCs and Müller glial cells. (**E**) UMAP of reclustered RPCs and Müller glial cells with colors indicating the sample origin of the cells. (**F**) Violin plots of area under the curve (AUC) scores of the collective gene expression levels of RPC feature gene set (left) and Müller feature gene set (right) in different clusters of reclustered RPCs and Müller glial cells. Clusters were ordered based on the scores. (**G**) UMAP graph illustrating the distribution of cell clusters of RPCs, RPCs committed to Müller glial fate (RPC_Müller), Müller glial cells just differentiated (Müller_Early), and Müller glial cells, as well as their corresponding mutant cell clusters (_mut). (**H**) Violin plot of AUC scores illustrating the activity of m⁶A-modified RPC genes in the control and mutant cell clusters along the course of RPC-to-Müller transition.

The online version of this article includes the following figure supplement(s) for figure 5:

**Figure supplement 1.** p7 retinal progenitor cells (RPCs) and Müller glial cells.

gene upregulation in *Mettl3*-mutant RPC and Müller glia clusters was due to lack of m⁶A-mediated degradation.

As in most neural tissues, gliogenesis occurs toward the end of retinogenesis. By constitutively expressing a set of progenitor genes, Müller cells maintain a certain level of retinal regeneration potential (*Blackshaw et al., 2004*; *Goldman, 2014*; *Jadhav et al., 2009*). However, to function as a glial cell, Müller cells need to establish a unique transcriptome different from that of progenitors (*Lin et al., 2019*; *Nelson et al., 2011*). How this transcriptomic transition from progenitors to glial cells is achieved remains unexplored. Taking advantage of our scRNA-seq data that likely contain transcriptomes of the cells in the transition states between late RPCs and Müller glial cells, we tested whether further analyses could identify such cell clusters. For this purpose, we pulled out late RPCs and Müller glial cells and reclustered them. Reclustering analyses generated nine clusters (*Figure 5D*). Cells originally annotated as RPCs and Müller cells were clearly separated on the two sides of the UMAP graph (*Figure 5—figure supplement 1A*), and cells in the inner circle of the graph were composed purely of cells from *Mettl3*-CKO retinas (*Figure 5E*), consistent with original clustering analyses (*Figure 3A, G and K*). In previous work, we compared the transcriptomes of Müller glia and early RPCs, and identified 3717 and 2450 genes that are significantly more abundant in early RPCs and Müller glia, respectively (*Lin et al., 2019*). Using these two groups of genes as the RPC signature gene set and Müller glia signature gene set, we used AUCell to score RPC and Müller glia features in each cell. Putting the two pure *Mettl3*-CKO cell cluster (clusters 1 and 6) aside, area under the curve (AUC) scoring showed that RPC clusters 2 and 7 exhibited significantly lower levels of the RPC feature but higher levels of the Müller feature than other RPC clusters, while Müller cluster 4 exhibited significantly lower level of the Müller feature but higher level of the RPC feature than other Müller clusters (*Figure 5F*), indicating that these three clusters reside in intermediate states between late RPCs and Müller glial cells. We thus classified cells in clusters 2 and 7 as late RPCs committed to Müller glial fate (named RPC_Müller), and cells in cluster 4 as Müller cells that had just exited the cell cycle and become Müller cells (named Müller_Early). The reclustering did not identify distinct mutant clusters corresponding to clusters 2, 4, and 7. Based on the expression patterns of markers for clusters 4, and 7, we classified subsets of cells of clusters 1 and 6 as the corresponding mutant clusters for clusters 4 and 7 (named Müller_Early_mut and RPC_Müller_mut, respectively); based on the pure *Mettl3*-CKO sample origin, we classified a subset of cells at one edge of cluster 2 as the corresponding mutant cluster for cluster 2 (also named RPC_Müller_mut). Thus, our reclusting and RPC/Müller feature analyses identified cell clusters in RPC-to-Müller transition states, as well as their corresponding mutant clusters (*Figure 5G*). We then scored the RPC and Müller features in this set of cell clusters. AUC scores showed that the RPC feature gradually decreased, and the Müller feature gradually increased from RPC clusters to Müller clusters in the control groups (*Figure 5—figure supplement 1E and F*). When examining the mutant groups, although the general trend was for the RPC feature to be elevated and for the Müller feature to be repressed compared with the corresponding control clusters, the differences were moderate (*Figure 5—figure supplement 1E and F*). We then determined the activities of subsets of RPC and Müller signature genes that carry m⁶A modification. Among the 3717 RPC signature genes, 1642 were subjected to m⁶A modification, and among the 2450 Müller signature genes, 894 were subjected to m⁶A modification. AUCell soring showed that, for the m⁶A subset of RPC signature genes, the activity in mutant clusters was significantly elevated to a level close to that in the control cluster of an earlier

state (*Figure 5H*), while for the m⁶A subset of Müller signature genes, the activity in the mutant clusters remained relatively unchanged (*Figure 5—figure supplement 1G*). Thus, these analyses demonstrated that m⁶A-RPC transcriptome failed to be promptly downregulated during the RPC-to-Müller transition.

Taken together, these analyses on the expression of m⁶A-tagged genes in control and *Mettl3*-mutant retinal cells demonstrate that m⁶A modification promotes the degradation of RPC transcripts during the RPC-to-Müller transition. This process is likely essential for the timely termination of retinogenesis and the proper establishment of the transcriptome of Müller glial cells.

## Overexpression of m⁶A-tagged RPC genes in late RPCs disturbs late-stage retinogenesis

To experimentally test whether m⁶A modification promotes the degradation of RPC transcripts, we selected eight m⁶A-tagged genes (*Zfp292*, *Ckap4*, *Nxt1*, *Apex1*, *Bcl7a*, *Traf4*, *Rap2b*, and *Sulf2*) that were significantly upregulated in the *Mettl3*-mutant RPC and Müller clusters (*Figure 6—figure supplement 1A*). All eight genes carry multiple m⁶A tags around the stop codon (*Figure 6—figure supplement 1B*) and are involved in various biological processes, such as transcription, cytoskeleton function, and nuclear export. Using MeRIP-qPCR, we confirmed that the m⁶A modification levels of the transcripts of these genes were reduced in *Mettl3*-CKO retinas (*Figure 6A*, *Figure 6—figure supplement 1C*). We next measured the half-lives of the transcripts of these genes in the control and *Mettl3*-CKO retinas at p1. The results showed that the half-lives of the transcripts of five of these genes were extended significantly in *Mettl3*-CKO retinas compared with those in the retinas of their littermate controls (*Figure 6B*, *Figure 6—figure supplement 1D*), demonstrating that m⁶A modification promotes the degradation of these RPC transcripts in the retina. To test how upregulation of m⁶A-regulated genes would affect late-stage retinogenesis, we performed in vivo electroporation to overexpress four genes that showed extended transcript half-lives in *Mettl3*-CKO retinas in p1 RPCs. At p3, 2 days after electroporation, we examined the cell cycle status of GFP-marked electroporated RPCs through KI67 or PCNA staining. The results showed that there were significantly fewer KI67⁻ or PCNA⁻; GFP⁺ cells in the retinas electroporated with *Zfp292*-, *Ckap4*-, *Traf4*-, and *Bcl7a*- overexpression plasmids (*Figure 6C and D*, *Figure 6—figure supplement 2A*), suggesting that overexpression of these four genes prohibited RPCs from exiting the cell cycle. After exiting the cell cycle, many differentiating retinal cells migrate away from the central region of the retinoblast layer to their designated position in the mature retina. In the control retinas, we observed many GFP⁺ cells located near the vitreous edge of the retinoblast layer, away from KI67-positive zones, indicating that they were differentiating/maturing amacrine cells (arrowheads in *Figure 6C*, Ctrl). In contrast, much fewer such cells were observed in retinas overexpressing *Zfp292*, *Ckap4*, *Traf4*, or *Bcl7a* (*Figure 6C*), further supporting the conclusion that overexpression of these four genes prohibited RPCs from exiting the cell cycle. We then traced the cell fates of GFP-marked electroporated RPCs at p14. Overexpression of *Zfp292*, *Ckap4*, *Traf4*, or *Bcl7a* led to an increase in the percentage of Müller cells at the expense of amacrine cells or rod photoreceptors (*Figure 6E and F*), suggesting that overexpression of these four genes biased the late RPCs to adopt Müller glial fate versus neuronal fates. Overexpression of *Rap2b* increased the population of bipolar cells and decreased the population of amacrine cells (*Figure 6—figure supplement 2B*). We then coelectroporated *Zfp292*, *Ckap4*, *Traf4,* and *Bcl7a* in the retina. Co-overexpressing of all four genes did not exhibit an additive effect (*Figure 6—figure supplement 2C*), suggesting that co-overexpressing all four genes might elicit different gene-regulatory network changes that are different from those when expressed alone. Phalloidin staining of the electroporated retinas at p14 showed that while the OLMs of these retinas were grossly intact, occasionally at a few regions (three out of six retinas), the OLMs are broken, and a few rods escaped to the subretinal space (*Figure 6—figure supplement 2D*, arrows in the inset image indicate the escaped rods). We then tested the effects of knocking down these m⁶A-regulated genes in late RPCs. We electroporated plasmids expressing shRNAs targeting *Zfp292*, *Ckap4*, *Traf4*, and *Bcl7a* into p1 retinas. Late RPCs deficient for *Bcl7a* generated fewer Müller glial cells, while knocking down other three genes did not significantly affect retinogenesis (*Figure 6—figure supplement 2E and F*). Taken together, in vivo electroporation experiments demonstrated that expression levels of these m⁶A-regulated genes should normally be kept at relatively low levels and carefully controlled in late RPCs.

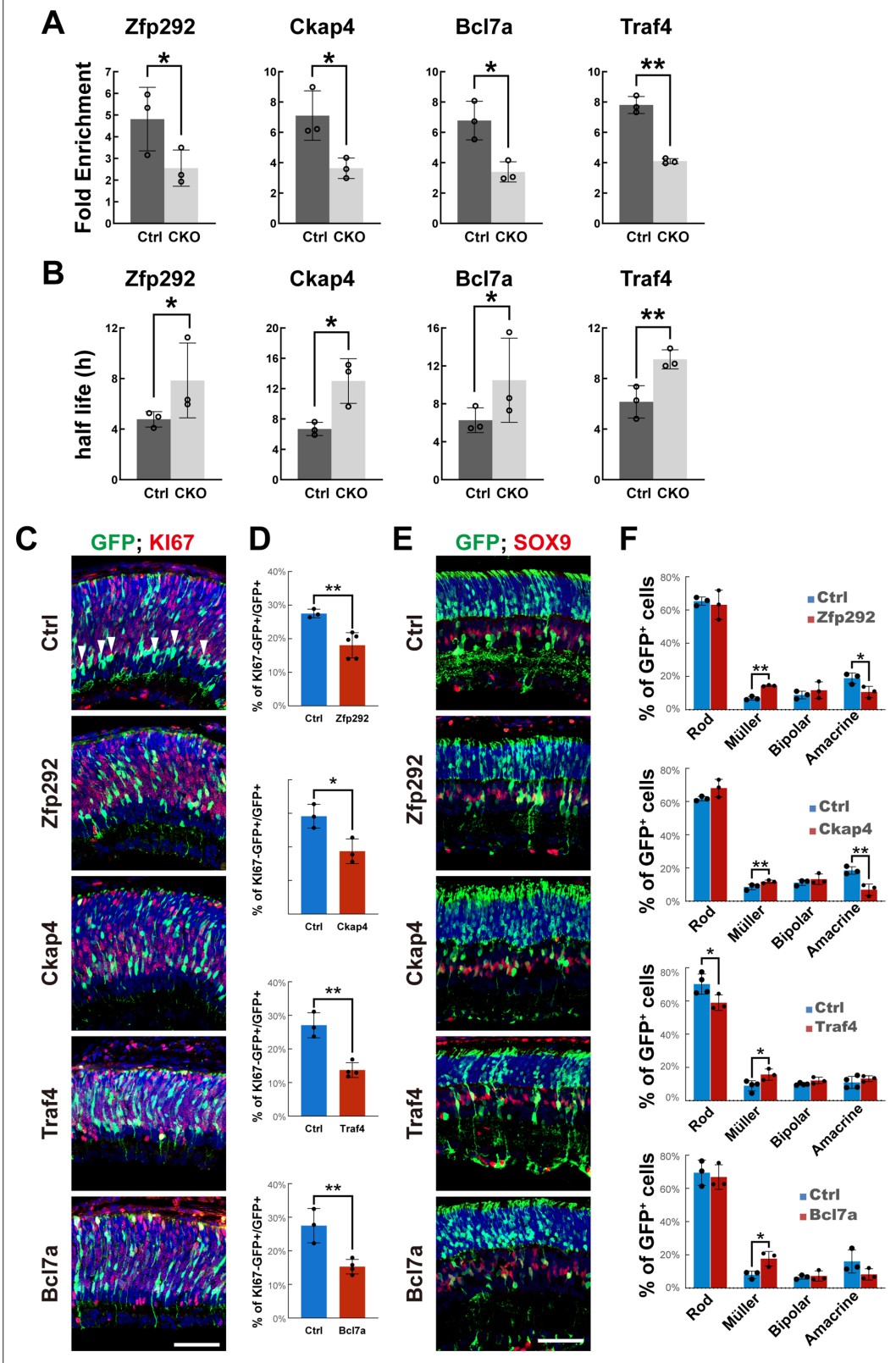

**Figure 6.** Overexpression of m⁶A-regulated genes disturbs late-stage retinogenesis. (**A**) MeRIP-qPCR results revealed downregulation of m⁶A modification of the transcripts in *Mettl3*-CKO retinas. (**B**) Transcript half-life measurements revealed prolonged half-lives of the transcripts in *Mettl3*-CKO retinas. (**C**) Confocal images of p3 retinas stained for GFP and KI67. The retinas were transfected with plasmids overexpressing cDNAs of candidate

*Figure 6 continued on next page*

*Figure 6 continued*

genes via in vivo electroporation at p1. (**D**) Quantification of the percentage of GFP-labeled electroporated RPCs that were KI67⁻ 48 hr after electroporation. (**E**) Confocal images of p14 retinas stained for GFP and SOX9. The retinas were transfected with plasmids overexpressing cDNAs of candidate genes via in vivo electroporation at p1. (**F**) Quantification of the cell compositions of GFP-labeled electroporated cells. The data in (**A, B, D, F**) are presented as the means ± standard deviation, corresponding to at least three independent biological replicates. *p<0.05, **p<0.01. The scale bars in (**C**) and (**E**) are 50 μm.

The online version of this article includes the following figure supplement(s) for figure 6:

**Figure supplement 1.** m⁶A-regulated genes.

**Figure supplement 2.** m⁶A-regulated genes control late-stage retinogenesis.

## Discussion

In this study, using *Mettl3*-CKO mice as a model, we revealed that *Mettl3* is essential for late-stage retinogenesis progression and structural and physiological homeostasis in the mature retina. Mechanistically, we showed that m⁶A promotes the degradation of RPC transcripts, thereby promoting the termination of retinogenesis and fine-tuning the transcriptomic transition from RPCs to Müller glial cells, the disruption of which leads to abnormalities in late-stage retinogenesis and compromises the glial functions of Müller cells and thus the structural and physiological homeostasis of the mature retina.

One interesting finding of our study is that the regulatory function of *Mettl3* during retinal development is restricted to the late-stage retinogenesis period. *Six3-Cre* mice express CRE recombinase in RPCs beginning from embryonic day 9.5 (*Furuta et al., 2000*); however, the retinas of *Mettl3*-CKO mice remained normal until the postnatal period. Our in vivo electroporation experiments to directly knock down *Mettl3* in late RPCs also demonstrated that *Mettl3* is required in late RPCs for the proper progression of late retinogenesis. Even though all are referred to as RPCs, RPCs continue to adjust their behavior during retinogenesis. For example, RPCs gradually go through a series of competent states to produce different types of retinal cells at different retinogenesis periods so that early RPCs produce retinal ganglion, cone, horizontal, and some amacrine cells, while late RPCs produce rod, some amacrine, and bipolar cells before finally becoming Müller glial cells (*Agathocleous and Harris, 2009*; *Cepko, 2014*). Accompanying the gradual changes in the retinogenic behaviors, RPCs likely continue to adjust the gene-regulatory networks to accommodate retinogenic tasks at different developmental stages. Indeed, scRNA-seq analyses of retinal development showed that the transcriptomes of early and late RPCs are distinct from each other (*Clark et al., 2019*). The postnatal retinogenesis period is the critical time when late RPCs quickly slow the cell cycle and finally exit the cell cycle to terminate retinogenesis in a timely fashion. Our study suggests that the m⁶A epitranscriptome delicately coordinates the transcriptomic adjustment needed for the timely termination of retinogenesis. In *Mettl3*-mutant RPCs, key cell cycle machinery components, such as *Ki67*, were upregulated, but many other cell cycle regulators were downregulated, which is likely the molecular basis for the distorted cell cycle. Furthermore, many m⁶A-tagged RPC transcripts were abnormally upregulated in *Mettl3*-mutant RPCs, which likely disrupts the balance of the gene-regulatory networks in late RPCs. Indeed, overexpression of m⁶A-tagged genes in late RPCs distorts the retinogenesis behavior of the cells. Thus, our study suggests that m⁶A delicately modulates the transcriptome of late RPCs to prepare them for the timely termination of retinogenesis.

Numerous studies have demonstrated the essential roles of Müller glial cells in maintaining the structural and physiological homeostasis of the mature retina (*Byrne et al., 2013*; *MacDonald et al., 2017*; *Shen et al., 2012*; *Wohl et al., 2017*). A couple of lines of evidence indicate that dysfunction of mutant Müller glial cells is the cellular basis for the retinal structural abnormality of *Mettl3*-CKO retinas. First, the structural abnormality of *Mettl3*-CKO retinas did not appear until p6, coinciding with the time point when maturing Müller glial cells are transitioning from the simple bipolar radial shape inherited from RPCs to a much more sophisticated morphology to form a glial network to intimately interact with every cell in the mature retina (*Wang et al., 2017*). Second, our scRNA-seq analyses demonstrated that Müller glia are one of the two cell types whose transcriptomes were most severely affected by *Mettl3* knockout, and many genes involved in cell–cell interaction and the cytoskeleton were downregulated in *Mettl3*-CKO Müller glial cells, suggesting that the structural

supporting function of Müller glial cells was compromised. On the other hand, late RPCs might indirectly contribute to the structural defect by failing to downregulate m⁶A-tagged transcripts, leaving a distorted transcriptome to their Müller progenies and compromised the glial supporting function of Müller cells, leading to the structural abnormality of the retina.

As the last cell type generated by RPCs at the end of retinogenesis and as the major glial cell type in the retina, the close relationship between Müller glia and RPCs has long been noted. Many factors that play important roles in RPC proliferation and differentiation are also highly expressed in Müller glia, and several transcriptomic comparison studies have revealed similarities between Müller glia and RPCs at the transcriptome level (*Blackshaw et al., 2004*; *Nelson et al., 2011*; *Roesch et al., 2008*). Encouraged by these cues, the regenerative potential of Müller glia has been intensively explored in recent years (*Goldman, 2014*; *Hoang et al., 2020*; *Jorstad et al., 2017*; *Lahne et al., 2020*; *Yao et al., 2018*; *Zhou et al., 2020*). However, to function as glial cells, Müller cells must develop a unique transcriptome that differs from that of RPCs to accommodate their structural and physiological supporting roles in the adult neural retina (*Lin et al., 2019*; *Nelson et al., 2011*), but how this delicate transcriptome transition is achieved is unclear. By carefully analyzing the single-cell transcriptomes of late RPCs and Müller glial cells of the p7 retina, we were able to capture cell clusters in RPC-to-Müller transition states. The RPC feature gradually decreases, and the Müller feature gradually increases, from RPCs to Müller glial cells through these clusters. However, in *Mettl3*-mutant cell clusters, the activities of m⁶A-tagged RPC genes were significantly elevated to a level close to those in the cluster of an earlier state. These analyses demonstrated that m⁶A promotes the transcriptomic transition from RPCs to Müller glial cells by promoting the degradation of modified RPC transcripts.

In our effort to test the roles of m⁶A-regulated genes in late retinogenesis, we used in vivo electroporation on retinas of newborn pups to either overexpress or knock down candidate genes. Knocking down candidate genes mostly did not affect late retinogenesis, while overexpressing some candidate genes disrupted the cell cycle progression of late RPCs and promoted Müller gliogenesis at the expense of rod or amacrine cell generation, suggesting that the expression levels of these genes in late RPCs should normally be kept at a relatively low level and carefully controlled. Overexpression of m⁶A-regulated genes did not faithfully recapitulate the cell population change phenotype observed in *Mettl3*-CKO retinas. For example, in *Mettl3*-CKO retinas, the population of rods decreased, while in retinas overexpressing *Zfp292* or *Ckap4*, the population of amacrine cells decreased. Two reasons might contribute to these discrepancies. First, in *Mettl3*-CKO RPCs, over 200 m⁶A-tagged genes were upregulated. The retinal cell-type composition change in *Mettl3*-CKO retinas is likely a collective result of the expression changes of many of these m⁶A-tagged genes; thus, overexpressing one (or a few) of them may not give rise to exactly the same phenotype as *Mettl3*-CKO retinas. Second, forced overexpression of *Zfp292*/*Ckap4*/*Traf4*/*Bcl7a* from plasmids electroporated into RPCs probably increases the expression levels of these genes by several to tens of times, while in *Mettl3*-CKO RPCs, their expression levels were only moderately elevated (*Source data 1 and 2*). Different levels of overexpression may elicit different gene-regulatory network changes, and thus different end results.

While this article was under revision, a manuscript was published showing that retinas deficient for all three m⁶A-readers, *Ythdf1*, *Ythdf2*, and *Ythdf3*, exhibit a phenotype similar to that of *Mettl3*-CKO retinas, suggesting that the three m⁶A readers work redundantly to 'read' m⁶A in the retina (*Niu et al., 2022*). As a derivative of the neural tube, the neural retina shares similar regulatory principles with other central neural tissues. Two groups independently studied the functions of m⁶A during cortical development using very similar strategies. Although the two studies presented controversial claims regarding the maintenance of neural stem cell (NSC) pools and the differentiation activities of NSCs, both studies showed that the cell cycle of NSCs deficient in *Mettl14* (another essential component of the m⁶A writer complex) was prolonged (*Wang et al., 2018*; *Yoon et al., 2017*), a phenotype shared by *Mettl3*-CKO RPCs, suggesting that some regulatory mechanisms of m⁶A are conserved between NSCs and RPCs. Similar to the case with Müller cells in the retina, the supporting functions and regeneration potentials of glial cells in the central nervous system are hotly debated neural biological questions (*Allen and Lyons, 2018*; *Guo et al., 2014*; *Qian et al., 2020*; *Wang et al., 2021*; *Zuchero and Barres, 2015*). Yoon et al. showed that there are fewer astrocytes in p5 *Mettl14*-deficient cortices than in control cortices; however, since neurogenesis is still progressing in *Mettl14*-deficient cortices at this time point, it is not clear whether the final population of astrocytes is reduced or increased in the adult *Mettl14*-CKO cortex. Furthermore, it is also not clear whether the generated *Mettl14*-deficient

astrocytes are transcriptomically and functionally normal (*Yoon et al., 2017*). Our study hints at an interesting possibility: that m⁶A may regulate the transcriptomic transition from NSCs to glial cells in other nervous systems, which warrants further investigation. Future studies are also needed to determine whether m⁶A is involved in cell fate reprogramming between glial cells and neurons.

# Materials and methods

**Key resources table**

| Reagent type (species) or resource | Designation | Source or reference | Identifiers | Additional information |
|---|---|---|---|---|
| Gene (*Mus musculus*) | *Mettl3* | GenBank | ID: 56335 | |
| Genetic reagent (*M. musculus*) | *Mettl3floxed* | **Lin et al., 2017** | | |
| Genetic reagent (*M. musculus*) | *Six3-Cre* | **Furuta et al., 2000** | | |
| Genetic reagent (*M. musculus*) | *Rlbp1-GFP* | **Vázquez-Chona et al., 2009** | | |
| Antibody | Anti-METTL3 (rabbit monoclonal) | Abcam | Cat# ab195352; RRID:AB_2721254 | 1:500 |
| Antibody | Anti-RECOVERIN (rabbit polyclonal) | Millipore | Cat# AB5585; RRID:AB_2253622 | 1:1000 |
| Antibody | Anti-VSX2 (sheep polyclonal) | Millipore | Cat# AB9014; RRID:AB_262173 | 1:500 |
| Antibody | Anti-SOX9 (rabbit polyclonal) | Millipore | Cat# ABS571; RRID:AB_2783876 | 1:500 |
| Antibody | Anti-CALBINDIN (rabbit polyclonal) | Swant | Cat# CB38; RRID:AB_10000340 | 1:500 |
| Antibody | Anti-SYNTAXIN (mouse monoclonal) | Sigma | Cat# S0664; RRID:AB_477483 | 1:200 |
| Antibody | Anti-BRN3A (mouse monoclonal) | Millipore | Cat# MAB1585; RRID:AB_94166 | 1:100 |
| Antibody | Anti-GFP (chicken polyclonal) | Abcam | Cat# ab13970; RRID:AB_300798 | 1:2000 |
| Antibody | Anti-N-CADHERIN (rabbit polyclonal) | Santa Cruz | Cat# sc-7939; RRID:AB_647794 | 1:1000 |
| Antibody | Anti-GS (mouse monoclonal) | BD Biosciences | Cat# 610518; RRID:AB_397880 | 1:2000 |
| Antibody | Anti-GFAP (mouse monoclonal) | Sigma | Cat# G3893; RRID:AB_477010 | 1:2000 |
| Antibody | Anti-BrdU (mouse monoclonal) | GE Healthcare | Cat# RPN202 | 1:50 |
| Antibody | Anti-pH3 (mouse monoclonal) | Cell Signaling | Cat# 9706; RRID:AB_331748 | 1:2000 |
| Antibody | Anti-KI67 (rabbit polyclonal) | Abcam | Cat# ab15580; RRID:AB_443209 | 1:500 |
| Antibody | Alexa Fluor 488 donkey anti-mouse (donkey polyclonal) | Thermo Fisher | Cat# A21202 | 1:1000 |
| Antibody | Alexa Fluor 568 donkey anti-mouse (donkey polyclonal) | Thermo Fisher | Cat# A10037 | 1:1000 |

*Continued on next page*

*Continued*

| Reagent type (species) or resource | Designation | Source or reference | Identifiers | Additional information |
|---|---|---|---|---|
| Antibody | Alexa Fluor 488 donkey anti-rabbit (donkey polyclonal) | Thermo Fisher | Cat# A21206 | 1:1000 |
| Antibody | Alexa Fluor 568 donkey anti-rabbit (donkey polyclonal) | Thermo Fisher | Cat# A10042 | 1:1000 |
| Antibody | Alexa Fluor 488 donkey anti-chicken (donkey polyclonal) | Jackson ImmunoResearch | Cat# 703-296-155 | 1:1000 |
| Recombinant DNA reagent | pCAGIG (plasmid) | Addgene | Cat# 11159 | CDS overexpression |
| Recombinant DNA reagent | pSicoR (plasmid) | Addgene | Cat# 11579 | shRNA expression |
| Sequence-based reagent | *Rsp14* forward | This paper | MeRIP-qPCR primers | acctggagcccagtcagccc |
| Sequence-based reagent | *Rsp14* reverse | This paper | MeRIP-qPCR primers | cacagacggcgaccacgacg |
| Sequence-based reagent | *Pax6* forward | This paper | MeRIP-qPCR primers | actctgccaatgactatgtg |
| Sequence-based reagent | *Pax6* reverse | This paper | MeRIP-qPCR primers | ctccagttcaggacagttac |
| Sequence-based reagent | *Sox2* forward | This paper | MeRIP-qPCR primers | ctggactgcgaactggagaa |
| Sequence-based reagent | *Sox2* reverse | This paper | MeRIP-qPCR primers | actctcctcttttttgcaccc |
| Sequence-based reagent | *Rax* forward | This paper | MeRIP-qPCR primers | ggaaattcagcctcgctgtc |
| Sequence-based reagent | *Rax* reverse | This paper | MeRIP-qPCR primers | ccaggtcaagatccttggtc |
| Sequence-based reagent | *Zfp292* forward | This paper | MeRIP-qPCR primers | gggaaataacgaatttcagg |
| Sequence-based reagent | *Zfp292* reverse | This paper | MeRIP-qPCR primers | ctctcttcaaattaccaggc |
| Sequence-based reagent | *Ckap4* forward | This paper | MeRIP-qPCR primers | ggaacgacctggataggttg |
| Sequence-based reagent | *Ckap4* reverse | This paper | MeRIP-qPCR primers | cgtaagaaactgtgcccacac |
| Sequence-based reagent | *Traf4* forward | This paper | MeRIP-qPCR primers | ggatgatgcggttttcatccg |
| Sequence-based reagent | *Traf4* reverse | This paper | MeRIP-qPCR primers | ccagtttcagatccagtcccg |
| Sequence-based reagent | *Bcl7a* forward | This paper | MeRIP-qPCR primers | ccaagaagaacctagagcgg |
| Sequence-based reagent | *Bcl7a* reverse | This paper | MeRIP-qPCR primers | ggcagtcacttgaaggttcg |
| Sequence-based reagent | *Rap2b* forward | This paper | MeRIP-qPCR primers | ggacttggagggtgaacgtga |
| Sequence-based reagent | *Rap2b* reverse | This paper | MeRIP-qPCR primers | gatgtctccatgaaggggcag |

*Continued on next page*

*Continued*

| Reagent type (species) or resource | Designation | Source or reference | Identifiers | Additional information |
|---|---|---|---|---|
| Sequence-based reagent | *Nxt1* forward | This paper | MeRIP-qPCR primers | ccagtaacacggtgtggaag |
| Sequence-based reagent | *Nxt1* reverse | This paper | MeRIP-qPCR primers | gagactggcatttctctgcag |
| Sequence-based reagent | *Apex1* forward | This paper | MeRIP-qPCR primers | gttgggatgaagccttcc |
| Sequence-based reagent | *Apex1* reverse | This paper | MeRIP-qPCR primers | catgagccacattgagatcc |
| Sequence-based reagent | *Sulf2* forward | This paper | MeRIP-qPCR primers | ctttggagaaagcacggac |
| Sequence-based reagent | *Sulf2* reverse | This paper | MeRIP-qPCR primers | cttctgagccagccaggtc |
| Other | *Mettl3* | GenBank | CGTCAGTATCTTGGGCAAATT | shRNA targeting site |
| Other | *Zfp292* | GenBank | TGTGGCAGTAAGCCATATATA | shRNA targeting site |
| Other | *Ckap4* | GenBank | ACGACCTGGATAGGTTGTTTC | shRNA targeting site |
| Other | *Traf4* | GenBank | GCGTATAGTTCCCACTAATTT | shRNA targeting site |
| Other | *Bcl7a* | GenBank | CGGAGCCAAAGGTTGATGATA | shRNA targeting site |

## Animals

All animal studies were performed in accordance with the protocol approved by the Institutional Animal Care and Use Committee of Zhongshan Ophthalmic Center (protocol number: Z2022011), and all animals were housed in the animal care facility of Zhongshan Ophthalmic Center *Six3-Cre* mice were kindly provided by Dr. Furuta from the University of Texas (**Furuta et al., 2000**). *Mettl3^floxed* mice were kindly provided by Dr. Tong Minghan from Shanghai Institute of Biochemistry and Cell Biology, Chinese Academy of Sciences (**Lin et al., 2017**). *Rlbp1-GFP* mice were kindly provided by Edward M. Levine from the University of Utah (**Vázquez-Chona et al., 2009**).

## Histology, immunostaining, and images

For cryosectioning, eyes were fixed overnight in 4% formaldehyde, cryopreserved with 15% and then 30% sucrose, frozen in Tissue-Tek OCT freezing medium, and sectioned using a cryostat microtome (Leica CM1950). For paraffin sectioning, eyes were fixed with Davidson's Fixative overnight, processed, embedded in paraffin, and sectioned using a microtome (Leica RM2235). Before subsequent histological staining, paraffin sections underwent a standard procedure for deparaffinization, and cryosections were air-dried for 5 min and then soaked in PBS for 5 min to wash off OCT. H&E staining was performed using a standard procedure. For fluorescent immunostaining, section slides were first heated in citrate buffer at 95°C for 30 min for antigen retrieval. After cooling to room temperature, slides were incubated with primary antibodies prepared in PBST with 5% fetal bovine serum at 4°C overnight, washed with PBST, incubated with fluorophore-conjugated secondary antibodies, washed, counterstained with 4',6-diamidino-2-phenylindole (DAPI), and finally mounted with cover glasses using VECTASHIELD mounting medium (Vector Labs). The following antibodies were used: rabbit anti-METTL3 (Abcam), rabbit anti-RECOVERIN (Millipore), sheep anti-VSX2 (Millipore), rabbit anti-SOX9 (Millipore), rabbit anti-CALBINDIN (Swant), HPC-1 (mouse anti-SYNTAXIN) (Sigma), mouse anti-BRN3A (Millipore), chicken anti-GFP (Abcam), rabbit anti-N-CADHERIN (Santa Cruz), mouse anti-GS (BD Transduction Laboratories), mouse anti-GFAP (Sigma), mouse anti-BrdU (Amersham), rabbit anti-KI67 (Abcam), mouse anti-pH3 (Ser10) (Cell Signaling Technology), Alexa Fluor 488 (568) donkey anti-mouse (Thermo Fisher), Alexa Fluor 488 (568) donkey anti-rabbit (Thermo Fisher), Alexa Fluor 488 donkey anti-chicken (Jackson ImmunoResearch), and Alexa Fluor 488 (568) donkey anti-sheep (Thermo Fisher). Bright-field H&E-stained images were taken under an inverted microscope (Zeiss Axio Observer). Immunofluorescence images were taken under a confocal microscope (Zeiss LSM 880 or LSM 980).

## BrdU labeling

Pups were injected subcutaneously with BrdU (0.32 μmol/g body weight). Then 2 hr, or 6 hr, or 48 hr later, pups were sacrificed, and eyes were collected, fixed with 4% formaldehyde overnight, and subjected to cryosectioning and immunofluorescent staining as described above. After confocal images were taken, the number of BrdU+ cells was counted for each selected 200 μm retinal region.

## TUNEL assay

Eye sections were prepared as described above for immunostaining. TUNEL assays were performed using an In Situ Cell Death Detection Kit (Roche) following the manufacturer's instructions.

## mfERG

The experiments were performed on p14 control (n = 4) and *Mettl3*-CKO (n = 2) mice. The mice were anesthetized using sodium pentobarbital and positioned on a warming table to maintain body temperature. For each animal, only the right eye was examined. Before recordings started, the eyes were dilated with 0.5% tropicamide (Santen Pharmaceutical, Japan) and local narcotized with 0.5% proparacaine hydrochloride (Ruinian Best, China). The eyes were positioned 1–2 mm in front of the device (Roland RETImap SLO, Roland, Germany). The animal Goldring recording electrode was placed at the corneal limbus. Subcutaneous silver needle electrodes served as reference and ground electrodes. An optical correction lens was positioned in front of the recording electrode. Viscous 2% methocel gel was applied between the cornea and contact lens. The stimuli were generated by a projector with a refresh rate of 60 Hz. The number of recording hexagons was set at 19, and the optic nerve head (ONH) was adjusted near the edge of the recording area. The stimulation parameter of light was set at mfERG Max. Twelve cycles were averaged for a final result.

## RNA-seq

RNA-seq was performed on four control and four CKO retinal samples. The retinas of p6 control and CKO pups were collected directly into TRIzol (Thermo Fisher), and total RNA was extracted following the manufacturer's instructions. Sequencing libraries were generated using the NEBNext Ultra RNA Library Prep Kit for Illumina (NEB) following the manufacturer's instructions. The libraries were sequenced on an Illumina HiSeq platform, and 150 bp paired-end reads were generated. Raw data were filtered to remove low-quality reads, and the clean reads were mapped to the mouse genome (GRCm38, mm10) using Hisat2 v2.0.5. featureCounts v1.5.0-p3 was used to count the read numbers mapped to each gene, and then the FPKM of each gene was calculated. Differential expression analysis was performed using the DESeq2 R package. The resulting p-values were adjusted using Benjamini and Hochberg's approach. Genes with an adjusted p-value<0.05 were considered differentially expressed.

## scRNA-seq

Two retinas from two control pups and two retinas from two *Mettl3*-CKO pups of the same litter were used for the scRNA-seq experiment. Retinas were digested with Papain (Worthington) to single cells and resuspended in PBS containing 0.04% BSA, and two retinal samples for each experimental group were mixed together as one sample for subsequent scRNA-seq. The scRNA-seq was performed by NovelBio Co., Ltd. Libraries were generated using the 10X Genomics Chromium Controller Instrument and Chromium Single Cell 3' V3.1 Reagent Kits (10X Genomics). Libraries were sequenced using an Illumina sequencer (Illumina) on a 150 bp paired-end run.

We applied fastp with default parameter filtering of the adaptor sequence and removed the low-quality reads to obtain clean data. Then, the feature-barcode matrices were obtained by aligning reads to the mouse genome (GRCm38, mm10) using CellRanger v3.1.0. Cells containing over 200 expressed genes and a mitochondrial UMI rate below 20% passed the quality filtering, and mitochondrial genes were removed from the expression table. The Seurat package v3.1 was used for cell normalization and regression to obtain the scaled data. PCA was constructed based on the scaled data with the top 2000 highly variable genes, and the top 10 principal components were used for UMAP construction. Utilizing the graph-based cluster method, we acquired unsupervised cell clusters. Judged by the key markers expressed by each cluster, we merged some clusters and annotated the final clusters.

For pseudotime analysis, we applied single-cell trajectory analysis using Monocle2 using DDR-Tree and default parameters. To identify DEGs among samples, the function FindMarkers with the Wilcox rank-sum test algorithm was used under the following criteria: lnFC > 0.25, p-value<0.05, min. pct > 0.1. To characterize the relative activity of gene sets with different m$^6$A features, we performed QuSAGE (2.16.1) analysis. To characterize the activities of RPC or Müller glial feature gene sets, R package AUCell was used. Fisher's exact test was applied to identify the significant GO terms and FDR was used to correct the p-values.

## MeRIP-seq

The retinas of C57BL/6J mice were collected directly into TRIzol (Thermo Fisher). Total RNA was extracted according to the manufacturer's instructions, and RNAs from 14 retinas were pooled together for each MeRIP-seq. MeRIP-seq was performed according to a published protocol (*Dominissini et al., 2013*). Briefly, PolyA-mRNA was purified using a Dynabeads mRNA DIRECT Purification kit (Thermo Fisher). mRNA was fragmented in fragmentation buffer heated at 94°C for 5 min. Fragmented mRNA was immunoprecipitated against an m$^6$A antibody (Synaptic Systems) bound to Protein G Dynabeads (Thermo Fisher) at 4°C for 2 hr, washed, and eluted by competition with free m$^6$A (Sigma). After recovery of precipitated m$^6$A-mRNA, sequencing libraries were constructed using the VAHTS Universal V6 RNA-seq Library Prep Kit (Vazyme). The libraries were sequenced on an Illumina HiSeq platform on a 150 bp paired-end run. Raw data were filtered to remove low-quality reads to obtain clean reads. Clean reads were mapped to the mouse genome (GRCm38, mm10) using BWA mem v0.7.12. m$^6$A peaks were determined using MACS2 v2.1.0 with the q-value threshold of enrichment set at 0.05. Motif analysis was done using MEME. GO term analysis was performed using clusterProfiler package (*Yu et al., 2012*).

## MeRIP-qPCR

Total RNA from retinas (5–6 retinas per sample) was extracted using TRIzol. MeRIP for qPCR was performed as described above for MeRIP-seq except that the mRNA fragmentation step was skipped. m$^6$A pull-down and 1% reserved input RNAs were reversed transcribed into cDNAs using the SuperScript II First-strand Synthesis System for RT-PCR (Thermo Fisher). qPCR was performed using the SYBR green-based method on a LightCycler 480 (Roche). The primers are listed in the key resources table. Enrichment of genes in m$^6$A-tagged genes in pull-down samples was calculated based on ΔCt between the pull-down sample and the input sample, and was normalized to an internal negative control, *Rsp14*, which does not carry m$^6$A modification.

## Molecular cloning

Full-length *Zfp292*, *Ckap4*, *Traf4*, *Bcl7a*, *Rap2b*, *Nxt1*, *Apex*, and *Sulf2* cDNAs were PCR amplified from a cDNA pool derived from embryonic mouse retina total RNA and cloned into the pCAGIG vector (gift from Dr. Connie Cepko and obtained through Addgene). shRNAs targeting *Zfp292*, *Ckap4*, *Traf4*, *Bcl7a*, and *Mettl3* were cloned into pSicoR vector (gift from Dr. Tyler Jacks and obtained through Addgene).

## Mouse retina in vivo electroporation

Newborn mouse pups were anesthetized by chilling on ice. A small incision was made in the eyelid and sclera near the edge of the cornea with a 30-gauge needle (Hamilton). Purified plasmid solutions (2–5 µg/µl) in PBS were injected into the subretinal space through the previous incision using a Hamilton syringe with a 32-gauge blunt-ended needle under a dissecting microscope. After DNA injection, a tweezer-type electrode was placed to softly hold the pup heads, and five 80 V pulses of 50 ms duration and 950 ms intervals were applied using a Gemini X2 pulse generator (BTX). After electroporation, pups were allowed to recover on a warming pad until they regained consciousness and were returned to their mother. The pups were sacrificed at p14, and the retinas were collected and subjected to cryosection as described above. Eye sections were immunostained for GFP (to recognize transduced cells and their progenies) and SOX9 (to recognize Müller cells). For cell composition analysis of the electroporated cells, GFP$^+$ cells in the ONL were judged as rods, GFP$^+$; SOX9$^+$ cells were judged as Müller cells, the GFP$^+$ cells in the INL above the SOX9$^+$ Müller cells (close to the OPL layer)

were judged as bipolar cells, while the GFP+ cells in the INL below the SOX9+ Müller cells (close to the IPL layer) were judged as amacrine cells.

## RNA half-life measurement

*Mettl3*-CKO and control mice were sacrificed at p1, and the retinas were collected and soaked in DMEM supplemented with actinomycin D (5 μm, Sigma) to inhibit transcription and cultured in a $CO_2$ incubator at 37°C. The retinas were collected directly into TRIzoL at 0 hr, 4 hr, 8 hr, and 16 hr after drug treatment. Total RNA was extracted according to the manufacturer's instructions. Two micrograms of total RNA from each sample was reverse transcribed into cDNA using SuperScript II (Thermo Fisher) and subjected to qPCR measurements using SYBR Green mix (Roche) and LightCycler LC480 (Roche). The primers are listed in the key resources table. The relative mRNA concentrations at each time point against time point 0 were calculated, the ln$^{\text{mRNA concentration}}$ at time points 0, 4, 8, and 16 hr were plotted to perform a linear regression analysis as a function of time, and the slope of the line was identified as the decay rate (k). The half-life was calculated with the following formula: t (1/2) = ln2/k (***Chen et al., 2008***).

## Statistical analysis

All experiments subjected to statistical analysis were performed with at least three biological replicates. Quantification data are presented as the mean ± standard deviation, and the significance of differences between samples was assessed using Student's *t*-test.

## Acknowledgements

We thank Dr. Minghan Tong, from Shanghai Institute of Biochemistry and Cell Biology, Chinese Academy of Sciences, for providing the *Mettl3*$^{\text{floxed}}$ mice, Dr. Edward M Levine, from the University of Utah, for providing the *Rlbp1-GFP* mice, Dr. Yasuhide Furata, from the University of Texas, for providing the *Six3-Cre* mice. We thank the staff of Laboratory Animal Center at State Key Laboratory of Ophthalmology, Zhongshan Ophthalmic Center for technical support. This study was supported by the National Natural Science Foundation of China (81870659 and 81721003).

## Additional information

### Funding

| Funder | Grant reference number | Author |
| --- | --- | --- |
| National Natural Science Foundation of China | 81870659 | Shuyi Chen |
| National Natural Science Foundation of China | 81721003 | Yizhi Liu |

The funders had no role in study design, data collection and interpretation, or the decision to submit the work for publication.

### Author contributions

Yanling Xin, Formal analysis, Investigation; Qinghai He, Huilin Liang, Ke Zhang, Jingyi Guo, Qi Zhong, Dan Chen, Jinyan Li, Investigation; Yizhi Liu, Conceptualization; Shuyi Chen, Conceptualization, Formal analysis, Supervision, Funding acquisition, Writing - original draft, Project administration

### Author ORCIDs

Shuyi Chen http://orcid.org/0000-0003-3037-6853

### Ethics

All animal studies were performed in accordance with the protocol approved by the Institutional Animal Care and Use Committee of Zhongshan Ophthalmic Center (protocol number: Z2022011) , and all animals were housed in the animal care facility of Zhongshan Ophthalmic Center.

Decision letter and Author response
Decision letter https://doi.org/10.7554/eLife.79994.sa1
Author response https://doi.org/10.7554/eLife.79994.sa2

## Additional files

### Supplementary files
- MDAR checklist
- Source data 1. List of differentially expressed genes (DEGs) between control and mutant retinal progenitor cell (RPC) clusters.
- Source data 2. List of differentially expressed genes (DEGs) between control and mutant Müller glial clusters.
- Source data 3. List of m⁶A peaks detected by MeRIP-seq in the retina.
- Source data 4. List of GO terms enriched in retinal m⁶A epitranscriptomes.

### Data availability
Sequencing data have been deposited in GEO under accession codes GSE180815, GSE181095, and GSE188607.

The following datasets were generated:

| Author(s) | Year | Dataset title | Dataset URL | Database and Identifier |
|---|---|---|---|---|
| Chen S | 2021 | Mettl3 regulates late retinogenesis | https://www.ncbi.nlm.nih.gov/geo/query/acc.cgi?acc=GSE181095 | NCBI Gene Expression Omnibus, GSE181095 |
| Chen S | 2021 | Mettl3 regulates late-stage retinogenesis and mouse retina m6A epitranscriptome | https://www.ncbi.nlm.nih.gov/geo/query/acc.cgi?acc=GSE180815 | NCBI Gene Expression Omnibus, GSE180815 |
| Chen S | 2021 | Neonatal and adult mouse retina m6A epitranscriptome [meRIP-seq] | https://www.ncbi.nlm.nih.gov/geo/query/acc.cgi?acc=GSE188607 | NCBI Gene Expression Omnibus, GSE188607 |

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
