## [Editor Report]

The authors' study investigated the role of m6A epitranscriptomic modification in the developing mouse retina. The study clearly demonstrated the defects of Mettl3CKO retina in mice, including cellular disorganization and abnormal physiological responses. Enriched scRNA-seq and MeRIP-seq data provide excellent resources to study the function of m6A modification in retinogenesis.

---

## [Decision Letter]

**Decision letter after peer review:**

Thank you for submitting your article "m^6^A epitranscriptomic modification regulates neural progenitor-to-glial cell transition in the retina" for consideration by *eLife*. Your article has been reviewed by 2 peer reviewers, and the evaluation has been overseen by a Reviewing Editor and Marianne Bronner as the Senior Editor. The reviewers have opted to remain anonymous.

Essential revisions:

1) The authors showed disrupted retina structure in Mettl3cko and claimed that dysfunction of Muller glial cells was the underlying cause. However, based on the data presented in the manuscript, it is not clear whether retinal structure abnormalities are direct effects of Muller glial defects or secondary effects induced by defects in RPCs or other cell types.

2) The authors claimed that m6A is important for RPC to Muller glial transition but didn't clearly state whether m6A regulates gliogenesis, which can be easily assessed by quantifying the number of Muller glial cells. The authors showed that the number of Muller glial cells was increased in mettl3cko at P7 based on *Sox9* staining, but the number of Muller glial cells in single cell RNA-seq data stayed unchanged between Mettl3cko and controls. In addition, the number of Muller glial cells in adult retinas (for example, at P14) was not examined. Based on images in Figure 1E, the number of Muller glial cells seemed to be slightly decreased compared to controls. Moreover, in line 130, the authors mentioned that Mettl3cko retinas resembled those with loss of Muller glia in the literature. These discrepancies need to be addressed.

3) The authors showed that there was an increased number of RPCs in Mettl3cko at late stages of retinal development compared to controls. However, some of the data do not seem to be consistent with published literature and their own supplementary data. Specifically, at P7, based on their single cell RNA-seq data, 6% of retinal cells were RPCs in control retinas, and 11% of retinal cells were RPCs in Mettl3cko (Figure 3—figure supplement 2). However, the number seems to be very high for control P7 retinas. In published single cell datasets (Clark et al. 2019), the percentage of retinal cells identified as RPCs was close to 0 at around P7. In addition, in Figure 3—figure supplement 3, the authors stained P7 Mettl3cko and control retinas with Ki67. There was a very limited number of Ki67+ cells in Mettl3cko, which doesn't quite match the 11% found in single cell RNA-seq data.

4) The stage of the retina was not consistent across experiments. The single cell RNA-seq experiment was performed using P7 retinas, whereas MeRIP-seq used P6 retinas. Twenty-four hours could make a big difference in terms of transcription at this stage.

5) Many claims in the manuscript are not fully supported by the data. For example, the authors over-expressed candidate m6A-regulated genes in RPCs and showed that they prevented the RPCs from exiting the cell cycle. However, the data presented in Figure 6 cannot fully support this conclusion. PCNA is not a pan cell cycle marker. Reduction of GFP+PCNA- cells in Mettl3cko doesn't necessarily mean that mutant cells failed to exit the cell cycle.

6) Individual data points and N numbers were not shown in the bar graphs.

7) The authors used six3:cre (expressed from E9.5) to knock out Mettl3 throughout the entire course of retinal development. It would be better if the authors could test the role of m6A in late RPCs directly.

8) In Figure 2, the authors identified müller glia in Mettl3CKO mice using the *Sox9* antibody. Due to severe structural abnormality of the retina in Mettl3CKO mice, the identification of cell fate needs additional caution. It is better to verify müller glia using other antibodies (such as glutamine synthetase, GS; GFAP) independently.

9) Some results seem inconsistent. First, in Mettl3CKO mice, the author found an increased number of Müller glia, a reduced number of rods, and no change in other cell types (Figure 2I-N). However, in the retinae with the overexpression of Zfp292, Ckap4, Traf4, or Bcl7a, they found an increase in the percentage of Müller cells at the expense of amacrine cells or rod photoreceptors (Figure 6C-D). Second, in scRNAseq data analysis, only the proportion of RPCs increased, but not Müller glial cells (Figure 3D). This result is not consistent with the IF phenotype (Figure 2I-J). The authors need to discuss these results at least to clarify the inconsistency.

10) Overall, the working model that the authors proposed lacks direct support. The authors may want to identify late RPC clusters that were committed to Müller glia fate as well as were subject to Mttl3CKO, and further verify the expression specificity of m6A-modified transcripts, such as Zfp292, Ckap4, Traf4, or Bcl7a, in such RPC clusters. Also, the authors should experimentally examine the essential roles of the loss of these m6A-modified transcripts in Müller glia development. These analyses will much strengthen the main conclusions of the study.

*Reviewer #1 (Recommendations for the authors):*

I encourage the authors to include the following info and additional explanations in the manuscript.

1. Line 120, the stage of IF staining should be included.

2. Line 146, please provide references showing that P6 is the peak time point for MG generation.

3. Line 162, the number of BrdU^+^;Ph3+ RPCs over BrdU^+^ cells should be quantified. The data presented by the authors cannot support the conclusion.

4. Line 180, please provide evidence showing that the major retinogenesis phase was completed in the central retinas in Mettl3CKO mice.

7. Line 183, how did the authors quantify Rho+ rods based on rhodopsin staining?

8. Line 226, "Considering that RPCs at this age are near the end of retinogenesis, most would become Müller glia. As a consequence, the final proportion of Müller glia in Mettl3CKO retinas should be higher than that in control retinas".

Muller glial proportion was not increased based on single cell RNA-seq data. This claim is confusing.

9. Line 258, "The upregulated genes were enriched for biological processes related to gene expression regulation, while downregulated genes were enriched for biological processes such as 'organelle organization' and 'precursor metabolites and energy'".

These GO terms do not seem to be the top terms based on the figures. Why did the author highlight them?

10. Line 259, "Importantly, key cell cycle machinery components, such as Ki67 and Brca2, was upregulated, but many cell cycle regulators, such as Ccna2 and Cdca8, were downregulated (Figure 3H-3J and Supplementary 261 Table 1), which explained why the cell cycle progression of Mettl3CKO RPCs was distorted".

This claim is very confusing. Does downregulation of Ccna2 and Cdca8 delay cell cycle?

11. Line 386, "p14 showed that while the OLMs of these retinas were grossly intact, at a few regions, the OLMs are broken, and a few rods escaped to the subretinal space".

This may be due to damage from retina sectioning. Please state whether this is a significant phenomenon. In addition, the authors did not quantify the number of Muller glial cells when the four plasmid mix was electroporated into P1 RPCs.

12. Figure 4 and Figure 5 may be combined.

*Reviewer #2 (Recommendations for the authors):*

1. The authors used six3:cre (expressed from E9.5) to knock out Mettl3 throughout the entire course of retinal development. It would be better if the authors could test the role of m6A in late RPCs directly.

2. In Figure 2, the authors identified müller glia in Mettl3CKO mice using the *Sox9* antibody. Due to severe structural abnormality of the retina in Mettl3CKO mice, the identification of cell fate needs additional caution. It is better to verify müller glia using other antibodies (such as glutamine synthetase, GS; GFAP) independently.

3. Some results seem inconsistent. First, in Mettl3CKO mice, the author found an increased number of Müller glia, a reduced number of rods, and no change in other cell types (Figure 2I-N). However, in the retinae with the overexpression of Zfp292, Ckap4, Traf4, or Bcl7a, they found an increase in the percentage of Müller cells at the expense of amacrine cells or rod photoreceptors (Figure 6C-D). Second, in scRNAseq data analysis, only the proportion of RPCs increased, but not Müller glial cells (Figure 3D). This result is not consistent with the IF phenotype (Figure 2I-J). The authors need to discuss these results at least to clarify the inconsistency.

4. Overall, the working model that the authors proposed lacks direct support. The authors may want to identify late RPC clusters that were committed to Müller glia fate as well as were subject to Mttl3CKO, and further verify the expression specificity of m6A-modified transcripts, such as Zfp292, Ckap4, Traf4, or Bcl7a, in such RPC clusters. Also, the authors should experimentally examine the essential roles of the loss of these m6A-modified transcripts in Müller glia development. These analyses will much strengthen the main conclusions of the study.

[Editors’ note: further revisions were suggested prior to acceptance, as described below.]

Thank you for resubmitting your work entitled "m^6^A epitranscriptomic modification regulates neural progenitor-to-glial cell transition in the retina" for further consideration by *eLife*. Your revised article has been evaluated by Marianne Bronner (Senior Editor) and a Reviewing Editor.

The manuscript has been improved but there are some remaining issues that need to be addressed, as outlined below:

While the reviewers appreciate the efforts that have been invested in this revised manuscript, they feel that an additional revision is needed to put the study into the context of previous report, and clarify the novelty of the findings.

*Reviewer #1 (Recommendations for the authors):*

The authors improved their manuscript. However, to make the study convincing, additional investigation and explanation still need to be conducted.

1. It is still not clear whether m6A depletion specifically affects RPC to muller glia (MG) transition, or just elongates the RPC cell cycle as shown in previously published papers, which investigated the role of m6A in brain development. Elongation of RPC cell cycle can also lead to increased production of MG.

2. The authors showed that there are significant cell death in the cko retinas at P0, P6 and P14, and this may be responsible for the not so significant increase of MG in cko. But they didn't show which types of cells are dying. Many TUNEL positive cells are in ONL (Figure 2-supplement 4, A'-C'). This cell death was also not carefully integrated into their conclusions when they explained the phenotypes or results.

3. The authors claimed that cko RPCs withdrew from the cell cycle shower than control RPCs (Figure 2E and F). If this is the case, there should be more proliferating RPCs in cko. However, the author showed that there are significantly less BrdU^+^ cells at P1 (2 hours after BrdU injection) in cko (Figure 2B). These results do not seem to support each other.

4. The authors didn't show that Mettl3 is depleted and m6A levels are lowered at embryonic stages.

5. For all the shRNA-based experiments, there aren't any control experiments to show their efficiency and specificity.

6. Lin 254-258, the argument is weak and hard to understand. The Ki67+ region is only at the very tip of the retina in cko. The authors need to provide stronger evidence to clarify whether the number of muller glial cells is increased or unchanged in the cko at different stages.

7. The authors claim that Mettl3cko "…compromises the function of Muller cells" (abstract line 32). There is no direct evidence to support this point.

8. The role of m6A on neural development has been extensively studied in the brain. The authors did not discuss these published papers and did not explain how their work improved the field.

Other concerns:

1. In Figure 1A and 1A', the background color looked very different. It looks like different exposure duration were used. A western blot may be better to show the deletion efficiency.

2. Penetrance (50%) of the OLM break phenotype was not included in the manuscript.

3. Line 161, 2 hour Brdu pulse chase experiment cannot support that the cells proliferate slower. It only suggests that there are less proliferating cells.

4. Figure 2D, label on Y axis was not corrected.

5. Line 201-206, the number of cells cannot be quantified based on markers expressed in the cytosol. These markers may be upregulated in individual cells.

6. The m6A IP was not shown for cko retinas in Figure 6-Supplement 1.

*Reviewer #2 (Recommendations for the authors):*

The authors have enlisted new experiments and analyses that substantially strengthen the paper in this revised manuscript. All these efforts should be applauded. In particular, the new data of shRNA targeting Mettl3 expressed in late RPCs directly addressed the cell-type specificity of Mettl3's role. Also, the detailed analysis of scRNA data of late RPCs further clarified the transition from late RPCs to the muller glial cells. Furthermore, the authors examined the influences of m6A-associated transcripts on muller glia development by electroporating shRNAs targeting these transcripts into P1 retinae. Besides, additional discussions in the revised manuscript help to clarify the concerns the reviewers raised. In all, all these improvements are satisfying. However, I am surprised that the authors did not provide the background of previous studies of m6A in retinal development. Adding these research backgrounds will further improve the clarity of this study by putting it into context.

---

## [Author Response]

Essential revisions:1) The authors showed disrupted retina structure in Mettl3cko and claimed that dysfunction of Muller glial cells was the underlying cause. However, based on the data presented in the manuscript, it is not clear whether retinal structure abnormalities are direct effects of Muller glial defects or secondary effects induced by defects in RPCs or other cell types.

We acknowledge that our study did not directly investigate what happens to the retina if *Mettl3* is specifically knocked out from Müller glial cells and their RPC precursors; we lack the necessary tools. However, several lines of evidence indicate that dysfunctional Müller glial cells are the key cells responsible for the structural abnormality of *Mettl3^CKO^* retinas:

1. As the sole neural glial cells inside the retina, Müller cells play key roles in maintaining the structural and physiological homeostasis of the mature retina, as demonstrated in numerous studies (MacDonald R et al., Curr Opin Neurobiol. 2017, 47:31-37; Shen W, et al., J Neurosci. 2012, 32(45):15715-27; Byrne L et al., PLoS One. 2013, 8(9):e76075; Wohl S et al., Nat Commun. 2017, 8(1): 1603).

2. The structural abnormality in *Mettl3^CKO^* retinas did not appear until p6, coinciding with the time point at which maturing Müller glial cells are transitioning from the simple bipolar radial shape inherited from RPCs to a much more sophisticated morphology; this morphology allows them to form a glial network and intimately interact with every cell in the mature retina (Wang J et al., J Comp Neurol. 2017, 525(8):1759-1777).

3. Our scRNA-seq analyses demonstrated that Müller glia were one of the two cell types whose transcriptomes were most severely affected by *Mettl3* knockout (the other being RPCs), and that many genes involved in cell-cell interactions and the cytoskeleton were downregulated in *Mettl3^CKO^* Müller glial cells, suggesting that the structural supporting function of Müller glial cells was compromised.

On the other hand, we did not exclude the involvement of other cell types, especially RPCs, in the structural defects. Indeed, we think that RPCs indirectly contributed to the structural abnormality: the transcriptome abnormality of *Mettl3^CKO^* Müller cells was rooted in the failure of RPC transcripts, especially m^6^A-tagged RPC transcripts, to be promptly downregulated in *Mettl3^CKO^* RPCs during RPC-to-Müller transition. The failure likely disrupted the balance of the gene-regulatory-network of the developing Müller glia and compromised the glial supporting function of Müller cells, leading to the structural abnormality of the retina.

We have included a discussion of the above in the Discussion section of the revised manuscript.

2) The authors claimed that m6A is important for RPC to Muller glial transition but didn't clearly state whether m6A regulates gliogenesis, which can be easily assessed by quantifying the number of Muller glial cells. The authors showed that the number of Muller glial cells was increased in mettl3cko at P7 based on Sox9 staining, but the number of Muller glial cells in single cell RNA-seq data stayed unchanged between Mettl3cko and controls. In addition, the number of Muller glial cells in adult retinas (for example, at P14) was not examined. Based on images in Figure 1E, the number of Muller glial cells seemed to be slightly decreased compared to controls. Moreover, in line 130, the authors mentioned that Mettl3cko retinas resembled those with loss of Muller glia in the literature. These discrepancies need to be addressed.

Regarding the number of Müller glial cells, at p7, more Müller glial cells were generated in the central retinas of *Mettl3^CKO^* mice than in those of control mice based on our manual counting of *SOX9*-stained retinal sections (We recounted these images, but the conclusion remains the same. Please see our response to comment #8). Then, why did the scRNA-seq analysis show that the percentage of Müller glial cells in *Mettl3^CKO^* retinas was comparable to that in control retinas at p7? This result is due to the different sampling strategies adopted in the manual image counting and scRNA-seq experiments: for manual counting, we selected the central region of the retina, while for scRNA-seq, we collected the whole retina. At p7, the remaining KI67^+^ retinogenesis region was larger in *Mettl3^CKO^* retinas than in control retinas (revised Figure 3—figure supplement 3). In other words, the region of the retina that had finished Müller gliogenesis was smaller in *Mettl3^CKO^* retinas than in control retinas. Thus, the elevated Müller gliogenesis phenotype was masked by the smaller region that had finished the process in the scRNA-seq data.

Regarding the population of Müller glial cells at p14, the reviewer is correct; the increased population of Müller cells in *Mettl3^CKO^* retinas observed at p7 was no longer obvious at p14. The reason for the declining population of Müller glial cells is the elevated cell death in *Mettl3^CKO^* retinas. Cell death was significantly elevated in *Mettl3^CKO^* retinas, including Müller glial cells (revised Figure 2—figure supplement 4), which led to the degeneration of all cell types in the retina, including Müller glial cells.

Regarding the term ‘gliogenesis’, our data clearly showed that more Müller glial cells were generated by *Mettl3^CKO^* RPCs than by control RPCs. However, we did not use the term ‘gliogenesis’ for two reasons: first, even though more Müller glial cells were generated by *Mettl3^CKO^* RPCs, the onset of Müller gliogenesis was delayed for *Mettl3^CKO^* RPCs; second, even though more Müller glial cells were generated by *Mettl3^CKO^* RPCs, the transcriptome of these cells was distorted with a certain level of RPC feature (revised Figure 5H), which likely compromised their glial function. For these reasons, we hesitate to say ‘m^6^A promotes gliogenesis’.

Regarding line 130, we have referenced the literature to emphasize that Müller glial cells play key roles in maintaining the structural homeostasis of the retina. In *Mettl3^CKO^* retinas, even though more Müller glial cells were generated than the number observed in control retinas, these cells are dysfunctional; thus, the retinas exhibited a phenotype similar to that of retinas in which Müller glial cells are depleted. We have revised the sentence for clarity.

3) The authors showed that there was an increased number of RPCs in Mettl3cko at late stages of retinal development compared to controls. However, some of the data do not seem to be consistent with published literature and their own supplementary data. Specifically, at P7, based on their single cell RNA-seq data, 6% of retinal cells were RPCs in control retinas, and 11% of retinal cells were RPCs in Mettl3cko (Figure 3—figure supplement 2). However, the number seems to be very high for control P7 retinas. In published single cell datasets (Clark et al. 2019), the percentage of retinal cells identified as RPCs was close to 0 at around P7. In addition, in Figure 3—figure supplement 3, the authors stained P7 Mettl3cko and control retinas with Ki67. There was a very limited number of Ki67+ cells in Mettl3cko, which doesn't quite match the 11% found in single cell RNA-seq data.

To address the reviewer’s concern, we applied the annotation criteria of Clark et al.’s 2019 Neuron paper to annotate our cells to determine how many cells would be annotated as late RPC. We used the R package SingleR and used p5 and p8 cells from the Neuron paper as reference cells to annotate our cells from control retinas. Please see the Author response image 1: 2.74% of our control cells were annotated as late RPCs.

**Author response image 1. sa2fig1:** 

We then pulled out data from Supplementary Table S5 of the Neuron paper and calculated the retinal cell type compositions in their p5 and p8 retinas. Please see Author response image 2: in the Neuron paper, at p5, 12.04% of retinal cells were annotated as late RPCs; at p8, 0.1% of retinal cells were annotated as late RPCs. They did not sequence p7 retinas.

Thus, for our p7 retinal cells, the percentage of cells annotated as RPCs in our paper (5.37%) (rounded to 6% by Excel, as shown in Supplementary Figure 8 in our original manuscript) is smaller than that annotated by SingleR based on the annotation standard in the Neuron paper (2.74%). However, after carefully inspecting the data, we feel that, at least for our p7 retinal cells, our annotation more closely reflects the real situation:1. For the other types of cells in p5, p8 and p14 retinas described in the Neuron paper, a few obviously abnormal numbers are evident (please see the Author response image 2): At p5, 20.04% of cells were annotated as Müller glial cells; the percentage fell to 0.48% at p8 but then increased to 17.86% at p14; the percentage of amacrine cells was 11.37% at p5, 0.26% at p8, and 0.14% at p14; and the percentage of bipolar cells was 1.18% at p5, 19.55% at p8, and 20.34% at p14. None of these numbers accurately reflect the real populations of cell types at the various time points, suggesting that caution should be taken when using these numbers.

2. We checked the cell cycle status of the cells annotated as “Müller Glia” in Clark et al.’s p5 retinas. Of 1087 cells, 68 were in G2/M phase, and 138 were in S phase; thus, approximately 19% of these cells were actually cycling RPCs, making the RPC percentage 15.9%.

3. Similarly, we checked the cell cycle status of our p7 retinal cells annotated as “Photoreceptor Precursor” by SingleR; 24% of them were in S phase, indicating that they were actually cycling RPCs.

4. The cell cycle status of the cells annotated as RPCs in our paper is either S phase or G2/M phase, demonstrating that these cells are proliferating RPCs, while most cells annotated as Müller cells were in G1 phase (Figure 5—figure supplement 1B).

5. Regarding the Mki67^+^ cells, as shown in Figure 5—figure supplement 1B-1D, only portion of proliferating RPC cells (mostly G1/M-phase cells) are Mki67^+^ cells. We calculated the percentages of Mki67^+^ cells in our scRNA-Seq data, which were 3.99% in control retinas and 7.93% in *Mettl3^CKO^* retinas.

6. To further address the reviewer’s concern about the discrepancy between the scRNA-seq data and the KI67 staining in our original Supplementary Figure 3, we performed KI67 staining on a few more p7 retinal samples and replaced the original images with images showing the whole retina. The abundances of KI67^+^ cells in these images roughly match the percentages calculated from the scRNA-seq data mentioned above.

Based on the above evidence, we feel that, at least for our p7 retinal cells, our annotation is close to the real situation.

4) The stage of the retina was not consistent across experiments. The single cell RNA-seq experiment was performed using P7 retinas, whereas MeRIP-seq used P6 retinas. Twenty-four hours could make a big difference in terms of transcription at this stage.

We performed p6 retina MeRIP-seq before scRNA-seq. Because the m^6^A epitranscriptome of the mouse retina is relatively stable during postnatal development (Figure 4A, and Figure 4—figure supplementary file 1 and file 2), we used p6 retina MeRIP-seq data in the scRNA-seq and MeRIP-seq integrative analyses in original manuscript. To address the reviewer’s concern, in the course of revising this manuscript, we collected p7 retinas and performed MeRIP-seq on them. Comparing the p7 MeRIP-seq results and p6 MeRIP-seq results showed that the two data sets were quite similar to each other (please see Supplementary Table 3 in the original manuscript and Figure 4—figure supplementary file 1 in the revised manuscript). We have replaced the p6 MeRIP-seq data with the p7 MeRIP-seq data in the revised manuscript.

5) Many claims in the manuscript are not fully supported by the data. For example, the authors over-expressed candidate m6A-regulated genes in RPCs and showed that they prevented the RPCs from exiting the cell cycle. However, the data presented in Figure 6 cannot fully support this conclusion. PCNA is not a pan cell cycle marker. Reduction of GFP+PCNA- cells in Mettl3cko doesn't necessarily mean that mutant cells failed to exit the cell cycle.

We agree with the reviewer that PCNA is not a pan cell cycle marker. To address the reviewer’s concern, we repeated these experiments with KI67. The results showed that, similar to the PCNA results, there were fewer GFP^+^; KI67^-^ cells among RPCs overexpressing *Zfp292*/*Ckap4*/*Traf4*/*Bcl7a* than in control RPCs. Although KI67 is not a pan cell cycle marker either (please see our response to comment #3), since both the PCNA- (S phase) and Ki67- (G2/M phase) cell populations were reduced in RPCs overexpressing *Zfp292*/*Ckap4*/*Traf4*/*Bcl7a*, the data strongly suggest that overexpressing *Zfp292*/*Ckap4*/*Traf4*/*Bcl7a* prevented RPCs from exiting the cell cycle.

Furthermore, another phenomenon indicates that fewer RPCs overexpressing *Zfp292*/*Ckap4*/*Traf4*/*Bcl7a* exited the cell cycle: after exiting the cell cycle, many differentiating retinal cells migrate away from the central region of the retinoblast layer to their designated position in the mature retina. For example, differentiating/maturing amacrine cells move toward the vitreous side of the retinoblast layer. In the control samples, we observed many GFP^+^ cells located near the vitreous edge of the retinoblast layer, away from Ki67- or PCNA- positive zones, indicating that they were differentiating/maturing amacrine cells (arrows in revised Figure 6C-Ctrl). In contrast, much fewer such cells were observed in retinas overexpressing *Zfp292*/*Ckap4*/*Traf4*/*Bcl7a*, suggesting that withdrawal from the cell cycle was delayed in these retina.

We have replaced the PCNA data in Figure 6 with the KI67 data and presented the PCNA data in Figure 6—figure supplement 2. We have also added a description of the tissue distribution patterns of GFP^+^ cells to the revised manuscript.

6) Individual data points and N numbers were not shown in the bar graphs.

We have added individual data points to the bar graphs. Since the N value is clearly reflected by the data points, we have not added N values to the bar graphs.

7) The authors used six3:cre (expressed from E9.5) to knock out Mettl3 throughout the entire course of retinal development. It would be better if the authors could test the role of m6A in late RPCs directly.

We thank the reviewer for the suggestion.

To test the role of m^6^A in late RPCs directly, we designed and synthesize plasmids expressing shRNAs targeting *Mettl3* and introduced these shRNAs into late RPCs through in vivo electroporation on retinas of p1 pups. We examined the cell proliferation status of the electroporated cells at p5. The results showed that, in the retinas electroporated with plasmids expressing shRNAs targeting *Mettl3*, significantly more GFP marked electroporated cells were labeled with 2-hour BrdU treatment than the retinas electroporated with the empty control vector (Figure 2—figure supplement 2), suggesting that cell cycle exit of the late RPCs in the experimental group was delayed. In addition, in the retinas electroporated with plasmids expressing shRNAs targeting *Mettl3*, fewer GFP marked electroporated cells had moved to the vitreous side of the retinoblast layer compared with the control retinas, indicating that the differentiation of *Mettl3* deficient late RPCs was delayed. Thus, these experiments demonstrated that *Mettl3* is required in late RPCs for their proper retinogenesis activities.

8) In Figure 2, the authors identified müller glia in Mettl3CKO mice using the Sox9 antibody. Due to severe structural abnormality of the retina in Mettl3CKO mice, the identification of cell fate needs additional caution. It is better to verify müller glia using other antibodies (such as glutamine synthetase, GS; GFAP) independently.

We agree with the reviewer that *SOX9* is not an ideal marker for Müller glial cells, especially for structurally distorted *Mettl3^CKO^* retinas. However, other Müller glial cell markers, such as GS and GFAP, have their own problems. GFAP is normally expressed in astrocyte but not Müller glial cells unless they are stimulated by injuries to the retina. GS is a cytoplasmic protein and presents challenges for counting Müller cells, which have complex cellular processes; moreover, at p7, GS is just starting to be expressed in Müller glial cells; the weak expression of GS in Müller glial cells (especially in *Mettl3^CKO^* retinas) at this time makes it unsuitable as a counting marker for Müller glial cells. To address the reviewer’s concern, we recounted *SOX9*^+^ cells in p7 retinas but only considered those located in the INL+OPL+ONL region (i.e., excluding those in the IPL+GCL region, which might be astrocytes) as Müller glial cells. The results showed that there were still more *SOX9*^+^ Müller glial cells in *Mettl3^CKO^* retinas than in control retinas (Revised Figure 2J). Thus, our original conclusion that more Müller glial cells were generated by *Mettl3^CKO^* RPCs than by control RPCs still stands.

9) Some results seem inconsistent. First, in Mettl3CKO mice, the author found an increased number of Müller glia, a reduced number of rods, and no change in other cell types (Figure 2I-N). However, in the retinae with the overexpression of Zfp292, Ckap4, Traf4, or Bcl7a, they found an increase in the percentage of Müller cells at the expense of amacrine cells or rod photoreceptors (Figure 6C-D). Second, in scRNAseq data analysis, only the proportion of RPCs increased, but not Müller glial cells (Figure 3D). This result is not consistent with the IF phenotype (Figure 2I-J). The authors need to discuss these results at least to clarify the inconsistency.

Regarding the reviewer’s concern about the discrepancy in Müller percentage numbers obtained from scRNA-seq analysis and IF image counting, please see our response to comment #2.

The following points address the reviewer’s concern that retinas overexpressing *Zfp292*/*Ckap4*/*Traf4*/*Bcl7a* did not exhibit exact the same cell composition changes observed for *Mettl3^CKO^* retinas: First, in *Mettl3^CKO^* RPCs, over 200 m^6^A-tagged genes were upregulated. It is likely that the change in retinal cell type composition in *Mettl3^CKO^* retinas was the collective result of expression changes of many of these m^6^A-tagged genes, and the overexpression of one (or a few) of them may not have given rise to the exact same phenotype as that of *Mettl3^CKO^* retinas. Second, forced overexpression of *Zfp292*/*Ckap4*/*Traf4*/*Bcl7a* from plasmids electroporated into RPCs (with CAG used as the promoter) probably increased the expression levels of these genes by several to tens of times, while in *Mettl3^CKO^* RPCs, the expression levels of these genes were only moderately elevated compared with those of control RPCs (Figure 3—figure supplementary file 1). Different levels of overexpression may elicit different changes in gene-regulatory-networks and thus yield different end results.

We have added the above information to the Discussion section of the revised manuscript.

10) Overall, the working model that the authors proposed lacks direct support. The authors may want to identify late RPC clusters that were committed to Müller glia fate as well as were subject to Mttl3CKO, and further verify the expression specificity of m6A-modified transcripts, such as Zfp292, Ckap4, Traf4, or Bcl7a, in such RPC clusters. Also, the authors should experimentally examine the essential roles of the loss of these m6A-modified transcripts in Müller glia development. These analyses will much strengthen the main conclusions of the study.

We thank the reviewer for the valuable suggestion.

To identify a putative RPC cluster that was committed to the Müller glia fate, we pulled out the data on RPC cells and Müller glial cells from the scRNA-Seq data (green and blue cells in Figure 3A) and reperformed clustering analyses, which separated the cells into 9 clusters. Clusters 0, 1, 3, and 4 were composed of cells originally annotated as Müller glial cells, while clusters 2, 5, 6, 7, and 8 were composed of cells originally annotated as RPCs (revised Figure 5D and Figure 5—figure supplement 1A). We then used AUCell and early RPC and Müller cell signature genes (Lin et al. Invest Ophthalmol Vis Sci. 2019, 60 (13): 4436-4450) to score the RPC and Müller features in each cell. Both the RPC feature scoring and the Müller feature scoring showed that RPC clusters 2 and 7 and Müller cell cluster 4 laid at the junction between RPCs and Müller cells (revised Figure 5F). We thus considered RPC clusters 2 and 7 as putative RPC clusters committed to the Müller glia fate and called them ‘RPC_Müller_’, and we considered Müller cell cluster 4 as the cluster that had just exited the cell cycle and become Müller cells and called them ‘Müller_Early_’. Our reclustering did not identify distinct mutant clusters corresponding to clusters 2, 4, and 7. Based on the expression patterns of markers for clusters 4 and 7, we classified subsets of cells of clusters 1 and 6 as corresponding mutant clusters for clusters 4 and 7 (‘Müller_Early_-mut’ and ‘RPC_Müller_-mut’), respectively. Based on the pure CKO sample origin, a subset of cluster 2 was considered the corresponding mutant cluster for cluster 2 (‘RPC_Müller_-mut’) (revised Figure 5G). Thus, by further analyses of RPCs and Müller cells, we identified an RPC cluster committed to the Müller glial fate and a Müller cluster at an early stage of differentiation, as well as their corresponding mutant clusters. We then used AUCell to score the expression levels of m^6^A-tagged RPC genes in the newly annotated cell clusters. The results showed that the expression levels of m^6^A-tagged RPC genes in mutant clusters were significantly elevated to a level close to that in the control cluster of an earlier state (Figure 5H). Thus, these analyses are consistent with our previous analyses (original Figure 5E), and strengthen our hypothesis that m^6^A promotes the transcriptomic transition from RPCs to Müller glial cells by promoting the degradation of modified RPC transcripts. We have added the above new analyses to the revised manuscript.

To examine the effects of the loss of these m^6^A-modified transcripts on Müller glia development, we designed and synthesized plasmids expressing shRNAs targeting *Zfp292*/*Ckap4*/*Traf4*/*Bcl7a* and electroporated these plasmids into p1 retinas. Late RPCs deficient for *Bcl7a* generated fewer Müller glial cells (rods and amacrine cells were slightly increased but did not show statistical significance), suggesting that *Bcl7a* promotes Müller glial cell development; while knocking down the other three genes did not significantly affect retinogenesis. Taken together, the gene knockdown and overexpression experiments suggest that the expression of these genes in late RPCs should normally be kept at low levels and carefully controlled. We have added the above new data to the revised manuscript.

Regarding the expression specificity of *Zfp292*, *Ckap4*, *Traf4*, and *Bcl7a*, AUC scoring of the expression levels of these genes in the above set of cell clusters showed that the expression levels of these genes were significantly elevated in all mutant clusters compared with the corresponding control clusters; however, within the control clusters, most of these genes did not show obvious RPC-specific expression (Figure 6—figure supplement 1A). We selected these m^6^A-regulated genes as RPC-specific genes based on our RNA-Seq comparison of sorted mature Müller glial cells and E12.5 retinas (Lin et al. Invest Ophthalmol Vis Sci. 2019, 60 (13): 4436-4450). Consistent with our RNA-seq comparison, analyzing Early RPC, Late RPC and Müller glia cells of Clark’s Neuron paper showed that these genes were more abundantly expressed in early RPCs, and often also so in late RPCs (please see Author response image 3). The lack of specificity of these genes in late RPCs at p7 might be due to the fact that, at this time point, the expression levels of these genes in this end-stage late RPCs were reduced to levels close to those in Müller glial cells. Because of the above new information, we feel that it is not accurate to call these genes late RPC-specific genes, but the results of overexpression and knockdown experiments targeting these genes were still in line with the conclusion that ‘m^6^A plays essential roles during late retinogenesis by promoting the degradation of m^6^A-regulated RPC transcripts’. We have revised the paragraph accordingly.

**Author response image 3. sa2fig3:** 

Reviewer #1 (Recommendations for the authors):I encourage the authors to include the following info and additional explanations in the manuscript.1. Line 120, the stage of IF staining should be included.

These experiments were performed on the retinas of p15 pups. We have added the information to the revised manuscript.

2. Line 146, please provide references showing that P6 is the peak time point for MG generation.

We intended to say ‘the time when morphological maturation of Müller glia starts’. The sentence has been revised and the corresponding reference has been added.

3. Line 162, the number of BrdU^+^;Ph3+ RPCs over BrdU^+^ cells should be quantified. The data presented by the authors cannot support the conclusion.

We have replaced Figure 2D with the quantification of BrdU^+^; pH3^+^ over BrdU^+^ percentages.

4. Line 180, please provide evidence showing that the major retinogenesis phase was completed in the central retinas in Mettl3CKO mice.

Please see Figure 3—figure supplement 3 in the revised manuscript, which showed that there were only a few KI67^+^ cells in the central retina of *Mettl3^CKO^* mice, suggesting that the major retinogenesis phase has passed in this region. We have replaced ‘completed’ with ‘passed’ in the revised manuscript.

7. Line 183, how did the authors quantify Rho+ rods based on rhodopsin staining?

We acknowledge that a cytoplasmic marker like RHODOPSIN is not ideal for counting cells, especially cells in the compacted ONL of the retina, and cones might be miss-counted as rods. We thus have revised the sentence to generally call RHODDPSIN^+^ cells as photoreceptors.

8. Line 226, "Considering that RPCs at this age are near the end of retinogenesis, most would become Müller glia. As a consequence, the final proportion of Müller glia in Mettl3CKO retinas should be higher than that in control retinas".Muller glial proportion was not increased based on single cell RNA-seq data. This claim is confusing.

We have revised the paragraph. Please see our response to comment #2.

9. Line 258, "The upregulated genes were enriched for biological processes related to gene expression regulation, while downregulated genes were enriched for biological processes such as 'organelle organization' and 'precursor metabolites and energy'".These GO terms do not seem to be the top terms based on the figures. Why did the author highlight them?

Among the 10 GO terms, 3 were related to organelles (organelle fission, organelle localization, organelle organization), 4 were related to molecule metabolism (precursor metabolites and energy, ATP metabolic process, ribonucleotide metabolic process, pyruvate metabolic process), and the remainder were related to the cell cycle. Our intent was to select one ‘organelle’ term and one ‘metabolism’ term as examples. We have revised the sentence.

10. Line 259, "Importantly, key cell cycle machinery components, such as Ki67 and Brca2, was upregulated, but many cell cycle regulators, such as Ccna2 and Cdca8, were downregulated (Figure 3H-3J and Supplementary 261 Table 1), which explained why the cell cycle progression of Mettl3CKO RPCs was distorted".This claim is very confusing. Does downregulation of Ccna2 and Cdca8 delay cell cycle?

Ccna2 plays an important role in regulating G1/S and G2/M transitions during the cell cycle; Cdca8 plays an important role in regulating mitosis. Thus, downregulation of these genes might be part of the molecular mechanism underlying the extended S-to-M phase progression observed for *Mettl3^CKO^* RPCs. We have revised the sentence.

11. Line 386, "p14 showed that while the OLMs of these retinas were grossly intact, at a few regions, the OLMs are broken, and a few rods escaped to the subretinal space".This may be due to damage from retina sectioning. Please state whether this is a significant phenomenon. In addition, the authors did not quantify the number of Muller glial cells when the four plasmid mix was electroporated into P1 RPCs.

We do not think the broken OLM in Figure 6E of the original manuscript was caused by tissue sectioning. Tissue sectioning could cause broken OLM, but in such situations, we do not see photoreceptors leak out as those shown in the DAPI channel insert image. However, we acknowledge that the penetrance of this phenotype was not high (3 out 6 retinas). We thus have moved these images to the supplementary figure and revised the text.

We have added the cell type composition quantification results of the retinas electroporated with the four plasmids mix in the revised manuscript (Figure 6—figure supplement 2C).

12. Figure 4 and Figure 5 may be combined.

In the revised manuscript, we have added new analyses to Figure 5 and moved some of the panels of Figure 5 to Figure 6. We feel it is better to keep Figure 4 and Figure 5 separate.

[Editors’ note: further revisions were suggested prior to acceptance, as described below.]

Reviewer #1 (Recommendations for the authors):The authors improved their manuscript. However, to make the study convincing, additional investigation and explanation still need to be conducted.Major concerns:1. It is still not clear whether m6A depletion specifically affects RPC to muller glia (MG) transition, or just elongates the RPC cell cycle as shown in previously published papers, which investigated the role of m6A in brain development. Elongation of RPC cell cycle can also lead to increased production of MG.

In Figure 5, we showed that *Mettl3*-mutant Müller cells retained certain features of RPCs, and the m^6^A-tagged RPC transcripts failed to be promptly downregulated during the RPC-to-Müller transition, demonstrating that m^6^A depletion directly regulates the RPC-to-Müller transition.

2. The authors showed that there are significant cell death in the cko retinas at P0, P6 and P14, and this may be responsible for the not so significant increase of MG in cko. But they didn't show which types of cells are dying. Many TUNEL positive cells are in ONL (Figure 2-supplement 4, A'-C'). This cell death was also not carefully integrated into their conclusions when they explained the phenotypes or results.

We performed costaining of retinal cell type markers with TUNEL, which showed that both RPCs and their differentiated progenies were dying. This led to the degeneration of the retina at older ages and might have contributed to the reduced population of photoreceptors at p7. We have added the above information to the revised version of the manuscript (Figure 2—figure supplement 4E-4G).

The elevations in cell death events would affect the cell composition of the retina, which is exactly why we chose p7 to count cells in the central retina, when the region had just finished retinogenesis, to keep the influence of the cell death events minimal. Why retinal cells die in *Mettl3*-CKO retinas is an interesting question that is worth future study. But, we believe further analyses on cell death will not affect the conclusions of this manuscript.

3. The authors claimed that cko RPCs withdrew from the cell cycle shower than control RPCs (Figure 2E and F). If this is the case, there should be more proliferating RPCs in cko. However, the author showed that there are significantly less BrdU^+^ cells at P1 (2 hours after BrdU injection) in cko (Figure 2B). These results do not seem to support each other.

Elevations in cell death activity (Figure 2—figure supplementary 4A’, 4D, and 4E) might reduce the population of RPCs in CKO retinas.

4. The authors didn't show that Mettl3 is depleted and m6A levels are lowered at embryonic stages.

We stained E15.5 retinas with a METTL3 antibody, which showed that *Mettl3* was efficiently deleted from the central retina (revised Figure2—figure supplement 1). Since no obvious phenotype was observed at this stage, we did not measure m^6^A level changes (which would require large amounts of tissues, meaning that large amounts of CKO and control embryos would need to be collected).

5. For all the shRNA-based experiments, there aren't any control experiments to show their efficiency and specificity.

We assessed the gene knockdown efficiencies of the shRNAs in mouse embryonic fibroblasts (MEFs): we generated shRNA-expressing lentiviruses (the cloning vector pSicoR is a lentivirus vector); used these viruses to infect MEFs; harvested MEFs two days after infection; and performed RT-qPCR (empty vector virus was used as control). We have added this information to the revised version of the manuscript (Figure6—figure supplement 2E).

6. Lin 254-258, the argument is weak and hard to understand. The Ki67+ region is only at the very tip of the retina in cko. The authors need to provide stronger evidence to clarify whether the number of muller glial cells is increased or unchanged in the cko at different stages.

Our manual counting of p7 retinas clearly demonstrated that more Müller cells were generated by CKO-RPCs. We have explained in detail why we chose p7 central retinas (see response to comment #2 above), how we counted (see response to comment #8 in the previous round of revision), why the increase in the population of Müller cells was not obvious at p14 (see response to comment #2 in the previous round of revision), and why the manual counting and scRNA-Seq results were seemingly inconsistent (see line 254-258, and the response to comment #2 in the previous round of revision). As we demonstrated in our response to comment #3 in our previous round of revision, bioinformatic analysis of scRNA-Seq data has its caveats and may not faithfully annotate every cell, but the general trends are usually consistent with the real situation in the tissue, which is the case here. We feel we have clearly demonstrated the point that more Müller cells were generated by CKO-RPCs.

7. The authors claim that Mettl3cko "…compromises the function of Muller cells" (abstract line 32). There is no direct evidence to support this point.

As we explained in our response to comment #1 in our previous round of revision, we acknowledge that our study did not directly test the role of *Mettl3* specifically in Müller cells, but several lines of our data indicate that CKO-Müller cells were dysfunctional. We have revised the sentence to “…likely compromises the glial functions of Müller cells.”.

8. The role of m6A on neural development has been extensively studied in the brain. The authors did not discuss these published papers and did not explain how their work improved the field.

Thank you for the suggestion.

In the Discussion section of the revised manuscript, we compare our study with two published studies investigating the roles of m^6^A during cortical development. Please see the last paragraph of the Discussion section of the revised manuscript.

Other concerns:1. In Figure 1A and 1A', the background color looked very different. It looks like different exposure duration were used. A western blot may be better to show the deletion efficiency.

We feel that our METTL3 immunostaining results clearly demonstrated that *Mettl3* was efficiently knocked out in the central region of the CKO retina at spatial and single-cell resolution (Figure 1A’), which is a better and more meaningful way than western blotting to demonstrate the deletion effect.

2. Penetrance (50%) of the OLM break phenotype was not included in the manuscript.

The information has been added to the revised manuscript.

3. Line 161, 2 hour Brdu pulse chase experiment cannot support that the cells proliferate slower. It only suggests that there are less proliferating cells.

We have revised the sentence to “…indicating that there were fewer proliferating RPCs in *Mettl3*-CKO retinas.”.

4. Figure 2D, label on Y axis was not corrected.

The error has been corrected.

5. Line 201-206, the number of cells cannot be quantified based on markers expressed in the cytosol. These markers may be upregulated in individual cells.

We agree with the reviewer that nuclear markers are better than cytosolic markers in terms of cell counting, but we think both can be used to count cells. We currently do not have suitable nuclear markers on hand for these experiments. The results may not be absolutely precise, but we believe they very closely approximate the real situation.

6. The m6A IP was not shown for cko retinas in Figure 6-Supplement 1.

We are not sure which panel this comment is referring to. If the reviewer is referring to panel B, we have the following response: Because MeRIP-Seq is not a good way to quantify differential methylation levels between samples (Mclntyre A, et al., Scientific Reports (2020), 10:590), we did not perform MeRIP-Seq for CKO retinas.

Reviewer #2 (Recommendations for the authors):The authors have enlisted new experiments and analyses that substantially strengthen the paper in this revised manuscript. All these efforts should be applauded. In particular, the new data of shRNA targeting Mettl3 expressed in late RPCs directly addressed the cell-type specificity of Mettl3's role. Also, the detailed analysis of scRNA data of late RPCs further clarified the transition from late RPCs to the muller glial cells. Furthermore, the authors examined the influences of m6A-associated transcripts on muller glia development by electroporating shRNAs targeting these transcripts into P1 retinae. Besides, additional discussions in the revised manuscript help to clarify the concerns the reviewers raised. In all, all these improvements are satisfying. However, I am surprised that the authors did not provide the background of previous studies of m6A in retinal development. Adding these research backgrounds will further improve the clarity of this study by putting it into context.

We are glad that the reviewer is satisfied with our revision efforts and thank you for the suggestion.

Before we started our *Mettl3*-CKO mouse project, we tested the function of m^6^A during vertebrate retinal development by knocking down m^6^A-writer components in zebrafish using morpholinos. The study was published last year in BBRC. In addition, during the time when this manuscript was under revision, A paper studying the role of m^6^A-readers during retinal development was published. We have discussed these works in the introduction or the Discussion sections in the revised manuscript.